# Natural Language PDDL (NL-PDDL) for Open-world Goal-oriented Commonsense Regression Planning in Embodied AI

**Xiaotian Liu**[1]*, **Armin Toroghi**[1]*, **Jiazhou Liang**[1], **David Courtis**[1], **Ruiwen Li**[1],
**Ali Pesaranghader**[2], **Jaehong Kim**[3], **Tanmana Sadhu**[2], **Hyejeong Jeon**[3], **Scott Sanner**[1,4]
[1]University of Toronto, [2]LG Electronics, Toronto AI Lab, [3]LG Electronics, [4]Vector Institute of AI
{xiaotian.liu,armin.toroghi,joe.liang,david.courtis}@mail.utoronto.ca,
ruiwen.li@alumni.utoronto.ca,
{ali.pesaranghader,jaehong02.kim,tanmana.sadhu,hyejeong.jeon}@lge.com,
ssanner@mie.utoronto.ca

## Abstract

Planning in open-world environments, where agents must act with partially observed states and incomplete knowledge, is a central challenge in embodied AI. Open-world planning involves not only sequencing actions but also determining what information the agent needs to sense to enable those actions. Existing approaches using Large Language Models (LLM) and Vision-Language Models (VLM) cannot reliably plan over long horizons and complex goals, where they often hallucinate and fail to reason causally over agent-environment interactions. Alternatively, classical PDDL planners offer correct and principled reasoning, but fail in open-world settings: they presuppose complete models and depend on exhaustive grounding over all objects, states, and actions; they cannot address misalignment between goal specifications (e.g., "heat the bread") and action specifications (e.g., "toast the bread"); and they do not generalize across modalities (e.g., text, vision). To address these core challenges: (i) we extend symbolic PDDL into a flexible natural language representation that we term NL-PDDL, improving accessibility for non-expert users as well as generalization over modalities; (ii) we generalize regression-style planning to NL-PDDL with commonsense entailment reasoning to determine what needs to be observed for goal achievement in partially-observed environments with potential goal–action specification misalignment; and (iii) we leverage the lifted specification of NL-PDDL to facilitate open-world planning that avoids exhaustive grounding and yields a time and space complexity independent of the number of ground objects, states, and actions. Our experiments in three diverse domains — classical Blocksworld and the embodied ALFWorld environment with both textual and visual states — show that NL-PDDL substantially outperforms existing baselines, is more robust to longer horizons and more complex goals, and generalizes across modalities.

## 1 Introduction

Open-world planning[1], where the agent operates under partial observability and incomplete knowledge, is essential for embodied agents to perform real-world tasks. Such embodied environments are inherently open-world and involve reasoning over a multitude of objects, posing challenges for planning. For example, we consider Figure 1, where the agent is asked to *"toast the bread and leave it on a plate"*. To efficiently identify the sequence of actions that achieve this goal given only partial observations of the state and an incomplete environment model, the agent must find the relevant

---

*Equal contribution

[1]In this work, we characterize *open-world* by partial observability and incomplete domain knowledge, distinguishing it from *open-world games*, which refers to open-ended tasks in expansive environments.

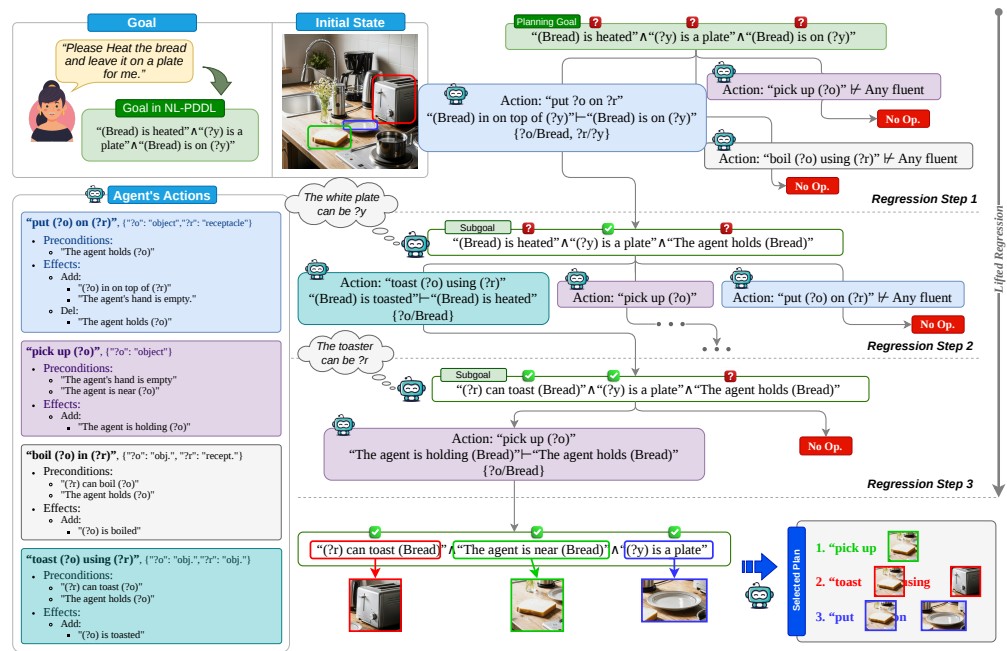

Figure 1: **Overview of the NL-PDDL framework:** (**Agent's Actions**) The specification of actions available to the agent are provided in a natural language (NL) variant of PDDL that we introduce. (**Goal**) The user provides NL instructions that are automatically translated into an NL-PDDL goal specification. (**Initial State**) The initial state is given as an image. (**Planning Goal**) Open-world first-order planning proceeds top-down: starting from the goal, regressed subgoals are generated that must hold before each action can be applied (e.g., the goal *(bread) is heated* regresses to the action *toast (bread) using (toaster)*, which generates subgoals *the agent holds the (bread)* and *a toaster is available*). At each step, LLM-based entailment connects subgoals with the NL effects of candidate actions and reasons about object affordances (e.g., *a toaster can toast bread, but a pot of water cannot*). (**Selected Plan**) A VLM grounds the NL object names in a regressed subgoal to their corresponding entities in the initial state image that are then used to instantiate an actionable plan.

objects (e.g., *bread*, *toaster*, and *plate*) while judiciously ignoring irrelevant ones. It must also use its commonsense knowledge to reason that a *toaster can heat bread*, but *a pot of water cannot*.

The breakthrough of foundation models such as Large Language Models (LLM) and Vision-Language Models (VLM) has led to their increased usage in planning (Yao et al., 2023b; Li et al., 2024; Kong et al., 2024); yet, they come with critical limitations (Shojaee et al., 2025; Kambhampati et al., 2024). These models struggle to generate reliable long-horizon plans with sound causal reasoning about the interactions between the agent and its environment as they lack a mechanism to track state changes and project how actions alter the world (Valmeekam et al., 2024). Moreover, they fail on complex goals involving multiple logical constraints (Goebel & Zips, 2025), and their black-box nature makes it hard to interpret or verify their generated plans (Aghzal et al., 2025). In contrast, classical planners, such as those based on the Planning Domain Definition Language (PDDL) (McDermott et al., 1998), have traditionally served the role of planning in AI. While verifiably correct, these methods often presume all objects and relations are known in a perfect model that can be exhaustively grounded. These methods also cannot bridge misalignments between goals and action specifications (e.g., reasoning that the action *"toast the bread"* is able to achieve the goal *"heat the bread"*a rev). Such limitations are too restrictive for the the open-world setting we address here.

To overcome the limitations of existing methods, we introduce NL-PDDL, a planning framework that combines the expressive flexibility of natural language (NL) with the formal guarantees of symbolic planning. An overview of the NL-PDDL workflow is provided in Figure 1 and its caption. Concretely, we make the following key contributions to open-world planning in embodied AI:

- We extend classical PDDL into NL-PDDL, a flexible representation that lets users specify goals and actions abstractly in NL. This flexibility reduces the specification burden (vs. rigid PDDL schemas) and tolerates semantically incomplete and syntactically imperfect domain and state descriptions while maintaining sound plan generation.

- We propose an open-world, regression-style NL-PDDL planner that uses LLM-based commonsense entailment over regressed subgoals to both infer the observations needed to achieve a target (e.g., *heat bread → find toaster*) and resolve misalignments between (sub)goals and action effects.

- We avoid exhaustive grounding via *lifted regression* (Reiter, 1991). Instead of reasoning about specific objects (e.g., *pan3*, *toaster1*), lifted regression leverages the variables inherent in NL-PDDL (e.g., *"there exists some x that can toast bread"*). Lifted plans are instantiated only when suitable objects are found, making planning complexity independent of the number of objects and abstractly capturing a lifted *conditional plan* of all initial state conditions that can achieve a goal.

- Across three diverse planning domains — Blocksworld (Valmeekam et al., 2022), ALFworld, and ALFworld-Vision (Shridhar et al., 2021) — we show that lifted regression planning in NL-PDDL achieves higher plan-success rates than strong baselines, remains robust as plan horizons increase, and generalizes across both text and vision modalities.

## 2 METHODOLOGY

In this section, we present Natural Language PDDL (**NL-PDDL**), a framework for open-world, goal-oriented planning by leveraging regression-style reasoning enhanced with LLM-based commonsense reasoning. While classical PDDL provides a sound framework for long-horizon planning, it cannot facilitate commonsense reasoning unless such knowledge is explicitly encoded by a domain designer. For example, a classical PDDL solver cannot infer that `isToasted(bread)` implies `isHeated(bread)`, even though it is intuitively clear to humans that *"if bread is toasted, it is likely heated"*. NL-PDDL builds on the formal framework of classical PDDL but replaces symbolic predicates, variables, and objects with typed NL counterparts, enabling seamless integration of LLMs to perform commonsense entailment inferences during planning. For a formal review of PDDL and first-order regression planning extended here, we refer the reader to Appendix B.

**Natural Language Representation of PDDL.** In classical PDDL, predicates, objects, and variables are represented with rigid symbols. NL-PDDL replaces these symbols with typed NL counterparts. For instance, the symbolic predicate `isToasted(bread)` becomes *"the (bread) is toasted"|"bread":"food"*, while *"the (?o:food) is toasted"* denotes its lifted form. Here, *?o:food* indicates that *(?o)* is of type *food*.

**NL-PDDL Problem Definition.** NL-PDDL problems are defined like classical PDDL problems, but aimed at open-world planning. We define an NL-PDDL problem as a tuple $\mathcal{P} = \langle G, \mathcal{A}, \mathcal{F}, h \rangle$, where $G$ is a conjunctive formula expressing the agent's goal, and $\mathcal{F}$ is the set of lifted NL predicates in the domain. $\mathcal{A}$ is the set of actions $a(\vec{y})$, i.e., parameterized first-order operators. Each $a(\vec{y}) \in \mathcal{A}$ is defined by: preconditions $a(\vec{y}).\text{pre} \subseteq \mathcal{F}$, which must hold for the action to execute; add effects $a(\vec{y}).\text{add} \subseteq \mathcal{F}$, which become true after the action is executed; and delete effects $a(\vec{y}).\text{del} \subseteq \mathcal{F}$, which no longer hold after execution. The planning horizon $h \in \mathbb{N}$ is the maximum length of feasible plans. The goal in NL-PDDL is to construct a *conditional plan* $\Pi$ that maps subgoal formulas to action sequences that can achieve them:

$$\Pi = \left[ (\psi_1(\vec{x_1}), \langle a^{1,1}(x^{\vec{1,1}}), ..., a^{1,n_1}(x^{\vec{1,n_1}}) \rangle), ..., (\psi_k(\vec{x_k}), \langle a^{k,1}(x^{\vec{k,1}}), ..., a^{k,n_k}(x^{\vec{k,n_k}}) \rangle) \right], \quad (1)$$

where each $\psi_i(\vec{x_i})$ is a first-order formula representing a subgoal, and $\langle a^{i,1}(x^{\vec{i,1}}), \ldots, a^{i,n_i}(x^{\vec{i,n_i}}) \rangle$ is the corresponding sequence of actions (i.e., a plan) with length $n_i \leqslant h$, and each $x^{\vec{j,k}} \subseteq \vec{x^j}$. As an example, the plan derived by NL-PDDL for the problem in Figure 1 is presented in Figure 2.

### 2.1 FIRST ORDER REGRESSION IN NL-PDDL

**Effect Axioms.** Following the methodology for first-order regression defined in Appendix B, we first need to transform our NL-PDDL description into the *effect axioms* that are used to perform regression planning. Specifically, following the notation of Appendix B, we build axioms of the form of Equations 8 and 9, and provide a concrete example of $\gamma^+_{p_{\text{toasted}}, a_{\text{toast}}}(?o, ?r)$ for action "toast

$$\Pi = [\,(\,\boxed{\psi_1 : \text{"(Bread) is heated"} \land \text{"(?y) is a plate"} \land \text{"(Bread) is on (?y)"}} \quad , \boxed{\langle \emptyset \rangle}\,),$$

$$(\,\boxed{\psi_2 : \text{"(Bread) is heated"} \land \text{"(?y) is a plate"} \land \text{"The agent holds (Bread)"}} \quad , \boxed{\langle \boxed{\text{"put (Bread) on (?y)"}} \rangle}\,),$$

$$(\,\boxed{\psi_3 : \text{"(?r) can toast (Bread)"} \land \text{"(?y) is a plate"} \land \text{"The agent holds (Bread)"}} \quad , \boxed{\langle \boxed{\text{"toast (Bread) using (?r)"}}, \boxed{\text{"put (Bread) on (?y)"}} \rangle}\,),$$

$$(\,\boxed{\psi_4 : \text{"(?r) can toast (Bread)"} \land \text{"(?y) is a plate"} \land \text{"The agent is near (Bread)"}} \quad , \boxed{\langle \boxed{\text{"pick up (Bread)"}}, \boxed{\text{"toast (Bread) using (?r)"}}, \boxed{\text{"put (Bread) on (?y)"}} \rangle}\,)\,]$$

Subgoals ⎵   Action Sequences ⎵

Figure 2: The conditional plan derived by NL-PDDL for the problem in Figure 1.

(?o:object) using (?r:object)" and predicate "(?o) is toasted" from the NL-PDDL description in Figure 3 [2].

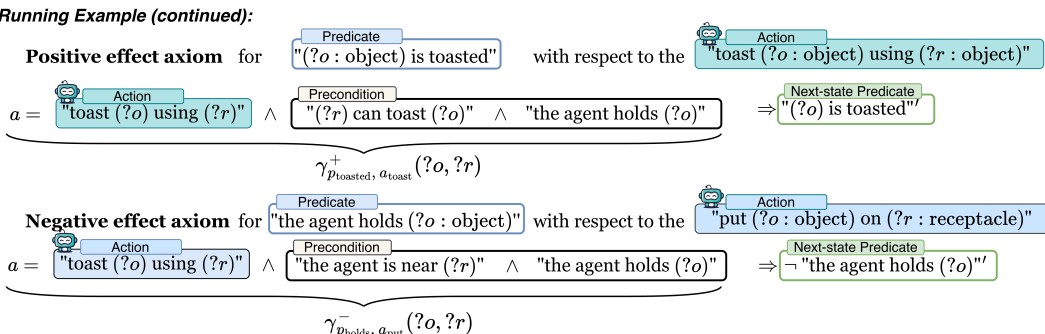

Figure 3: Example of positive and negative effect axioms based on the NL-PDDL problem in Figure 1 .

In English, this says an object *"(?o)"* is toasted in the next state if the agent holds it, and the agent toasts *"(?o)"* using object *"(?r)"* that can toast *"(?o)"*. To illustrate negative effect axioms, we construct $\gamma^-_{p_{\text{holds}}, a_{\text{put}}}(?o, ?r)$ for action *"put(?o) on (?r)"* and predicate *"the agent holds(?o)"* in Figure 3. In English, this says if the agent holds an object *"(?o)"*, is near receptacle *"(?r)"*, and puts *"(?o)"* on *"(?r)"*, then the agent no longer holds *"(?o)"* in the next state.

These positive (negative) effect axioms are only constructed for predicates and actions if the predicate is in the add (delete) effect of the action in the NL-PDDL description. Cases that a predicate appears in the effects of multiple actions are handled via disjunction (see Appendix C for details).

**Successor State Axioms.** As described in Appendix B, successor state axioms (SSAs) are the workhorse of lifted regression planning in that they allow the replacement of a predicate in a post-action state with the necessary subgoals required to achieve it. Let $F'_{a_i}(\vec{y_F})$ denote the value of a predicate $F(\vec{y_F})$ in the next state after executing action $a_i(\vec{y})$. The SSA for $F(\vec{y_F})$ is defined as:

$$F'_{a_i}(\vec{x}) \equiv \gamma^+_{F,a_i}(\vec{x}) \ \lor \ \left( F(\vec{x}) \land \neg\gamma^-_{F,a_i}(\vec{x}) \right). \tag{2}$$

In English, an SSA states that a predicate $F$ is true in the successor state *iff* it was made true by a positive effect, or it was already true and not made false by a negative effect. We construct the SSAs for each action–predicate pair. While we leave complete formal details to Appendix B due to space limitations, we provide the SSA for predicate $p_{\text{toasted}}(?o)$ and action $a_{\text{toast}}(?o, ?r)$, and for predicate $p_{\text{holds}}(?o)$ and action $a_{\text{put}}(?o, ?r)$ in Figure 4 as examples to illustrate the process of SSA. After constructing the SSAs, NL-PDDL recursively regresses each subgoal through applicable actions to construct the conditional plan $\Pi$. We explain the regression procedure in the next section.

### 2.1.1 REGRESSION OF POTENTIALLY MISALIGNED PREDICATES IN NL-PDDL

In regression planning with classical PDDL, goal predicates are only matched to action effects that share the same predicate names. In NL, such exact matches rarely occur due to variability in phras-

---

[2] All unquantified variables are assumed to be universally quantified.

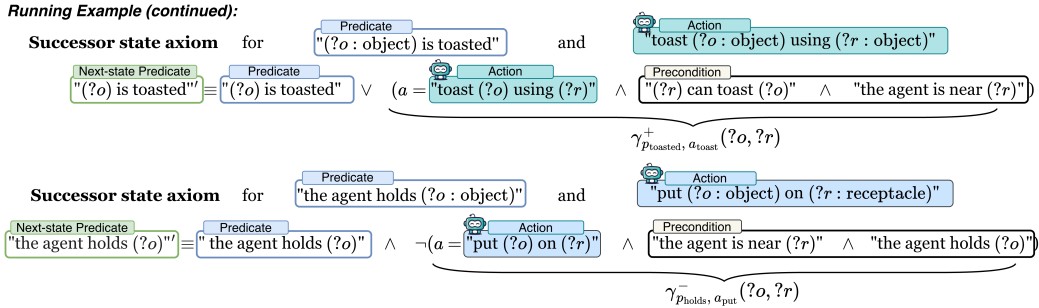

Figure 4: Example successor state axioms induced from the corresponding positive and negative effect axioms.

ing, instead an action effect may entail a goal literal, e.g., "the bread is toasted" entails ($\vdash$) "the bread is heated". In this section, we extend lifted regression planning using LLM entailment ($\vdash_{\text{LLM}}$).

**Regression of Positive NL Predicates.** Let $P'(\vec{x})|\vec{x}:\vec{T_x}$ be a predicate in the next state. We aim to regress it through an action $a_i$, to derive pre-action conditions such that if they hold, executing $a_i$ results in a next state that entails $P'(\vec{x})$. To this end, we first identify a predicate $F(\vec{z}) \in \mathcal{F}|\vec{z}:\vec{T_z}$ that can be unified with $P'(\vec{x})$. In PDDL, unification makes the two predicates syntactically identical through variable substitution, but NL-PDDL allows a *more general form of unification based on commonsense entailment*: predicates $F(\vec{z})$ and $P'(\vec{x})$ are unifiable *iff* they have equal arities and a substitution $\theta$ exists that induces an injective mapping between their variables such that:

**(i)** For every pair of corresponding types $t_z \in \vec{T_z}$ and $t_x \in \vec{T_x}$ mapped by $\theta$, $t_x \vdash_{\text{LLM}} t_z$.

**(ii)** After applying $\theta$, the substituted predicate $F(\vec{z})$ entails the substituted goal predicate $P'(\vec{x})$, i.e., $F(\vec{z})\theta \vdash_{\text{LLM}} P'(\vec{x})\theta$. [3]

A legal typed unification results in a substitution $\theta$ that is a mapping of variables to terms (e.g., $\theta = \{x/y\}$, so that $F(x)\theta \equiv F(y)$), which then permits a misaligned regression to proceed. Formally, let $F'_{a_i}(\vec{z}) \in \Phi$ denote the SSA formula for $F'(\vec{z})$ with respect to $a_i$ under the unifier $\theta$, then we can obtained the regressed condition for $P'(\vec{x})$ with respect to $a_i$ as proved in Appendix E:

$$\text{Regr}_\vdash\big(P'(\vec{x}), a_i\big) \equiv F'_{a_i}(z)\theta. \tag{3}$$

With $\text{Regr}_\vdash$, we can now formally regress a predicate through an action with misaligned effects:

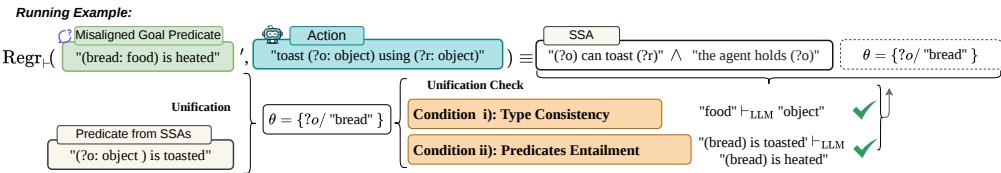

Figure 5: An example of a regression step of a positive predicate with goal–action misalignment.

In the example of Figure 5, to identify the subgoals necessary for the predicate *"(Bread:food) is heated"* to hold after an action is executed, we aim to find a predicate unifiable with it. The predicate *"(?o:object) is toasted"* qualifies, with $\theta = \{$*"?o"/"bread"*$\}$, because **(i)** type consistency holds (*"food"* $\vdash_{\text{LLM}}$ *"object"*), and **(ii)** after substitution, (*"bread is toasted"* $\vdash_{\text{LLM}}$ *"bread is heated"*). This allows us to derive the regressed subgoal condition (given by the SSA) that can achieve *"(bread) is heated"*, which is how subgoal $\psi_3$ is derived from $\psi_2$ in the conditional plan example of Figure 2.

---

[3]This relaxation from syntactic equivalence to entailment-based unification allows, for example, unifying *"(?x:food) is on (?y:plate)"* with *"(?y:receptacle) is under (?x:object)"*. After standardizing variables apart by renaming the second predicate to *"(?w:receptacle) is under (?z:object)"*, the substitution $\theta = \{?z/?x, ?w/?y\}$ aligns their arguments, yielding *"(?x) is on (?y)"* and *"(?y) is under (?x)"*. This unification is allowed since we have *"food"* $\vdash_{\text{LLM}}$ *"object"*, *"plate"* $\vdash_{\text{LLM}}$ *"receptacle"*, and *"(?x) is on (?y)"* $\vdash_{\text{LLM}}$ *"(?y) is under (?x)"*.

**Regression of Negative NL predicates.** As for negative predicates, we regress a negated predicate $\neg P'(\vec{x})|\vec{x}:\vec{T_x}'$ through an action $a_i$ to derive preconditions that ensure $\neg P'(\vec{x})$ holds after $a_i$ is executed. Thus, we identify a predicate $F(\vec{z}) \in \mathcal{F}|\vec{z}:\vec{T_z}$ that can be unified with $P'(\vec{x})$. Let $\theta$ be the unifier inducing an injective mapping between predicate variables such that:

   **(i)** For every pair of corresponding types $t_x \in \vec{T_x}$ and $t_z \in \vec{T_z}$ mapped by $\theta$, $t_x \vdash_{\text{LLM}} t_z$.

   **(ii)** After applying $\theta$, the substituted goal predicate entails the substituted precondition predicate, i.e., $P'(\vec{x})\theta \vdash_{\text{LLM}} F(\vec{z})\theta$.

Critically note that the direction of entailment in check (ii) for negative NL predicates reverses from the case for positive NL predicates. Let $F'_{a_i}(\vec{z}) \in \Phi$ be the SSA formula for $F'(\vec{z})$, with respect to $a_i$. Applying $\theta$ to this formula and negating the result gives the regressed condition for $\neg P'(\vec{x})$ with respect to $a_i$ as proved in Appendix E:

$$\text{Regr}_\vdash\big(\neg P'(\vec{x}), a_i\big) \equiv \neg F'_{a_i}(\vec{z})\theta. \tag{4}$$

Figure 6 illustrates a worked example of the regression of a negative NL predicate. To identify the subgoals necessary for the predicate $\neg$*"the agent possesses(bread:food)"* to hold after an action is executed, we aim to find a predicate that is unifiable with it. The predicate $\neg$*"the agent holds(?o:object)"* qualifies for this, under the substitution $\theta = \{$*"?o"/"bread"*$\}$, because **(i)** type consistency condition holds (*"food"* $\vdash_{\text{LLM}}$ *"object"*), and **(ii)** once the substitution is applied, (*"the agent possesses (bread)"* $\vdash_{\text{LLM}}$ *"the agent holds (bread)"*). Note that the direction of the entailment in condition **(ii)** is the opposite of the case of positive predicates. The post-action predicate $\neg$*"the agent possesses(bread:food)"* can thus be unified with the post-action predicate $\neg$*"the agent holds(?o:object)"*, which is in turn equivalent to its SSA under the action *"pick up (?o)"*, which yields the new subgoal.

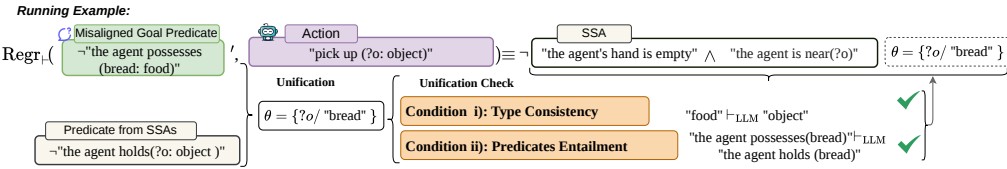

Figure 6: An example of a regression step of a negative predicate with goal–action misalignment.

**NL Regression with Multiple Entailments.** A predicate $P'(\vec{x})|\vec{x}:\vec{T}$ may be semantically entailed by multiple other predicates. For example, the goal *"(?o) is cooked"* may be entailed by either *"(?o) is toasted"* or *"(?o) is boiled"*. In such cases, we aggregate all entailing predicates. Let $F^+_\vdash = \{ F_1(\vec{x}_1), \ldots, F_m(\vec{x}_m) \mid F_i(\vec{x}_i) \in \mathcal{F} \ \wedge \ F_i(\vec{x}_i)\{\vec{x}_i/\vec{x}\} \vdash_{\text{LLM}} P'(\vec{x}) \}$ be the set of all predicates that entail $P'(\vec{x})$. The aggregated regression of $P'(\vec{x})$ with respect to action $a_i$ is:

$$\text{Regr}_\vdash\big(P'(\vec{x}), a_i\big) \equiv \bigvee_{j=1}^{m} F'_{a_i,j}(\vec{x}), \quad \text{where } F_j \in F^+_\vdash. \tag{5}$$

The same construction applies to negated predicates, with entailment checked in the reverse direction. Let $F^-_\vdash = \{ F_1(\vec{x}_1), \ldots, F_m(\vec{x}_m) \mid F_i(\vec{x}_i) \in \mathcal{F} \ \wedge \ P'(\vec{x})\{\vec{x}_i/\vec{x}\} \vdash_{\text{LLM}} F_i(\vec{x}_i) \}$ be the set of all predicates entailed by $P'(\vec{x})$. The aggregated regression of $\neg P'(\vec{x})$ with respect to $a_i$ is:

$$\text{Regr}_\vdash\big(\neg P'(\vec{x}), a_i\big) \equiv \bigvee_{j=1}^{m} \neg F'_{a_i,j}(\vec{x}), \quad \text{where } F'_j \in F^-_\vdash. \tag{6}$$

Equations 5 and 6 are proved in Appendix E. In practice, a large number of entailments can induce a cross-product explosion during DNF expansion, thus substantially prolonging regression time.

---

**Algorithm 1** REGRESSFORMULA($\psi'$, $a_i(\vec{y_i})$, $\mathcal{F}$)

1: **Input:** $\psi'$ in DNF; $a_i(\vec{y_i})$; $\mathcal{F}$
2: **Output:** $\psi = \text{Regr}_\vdash(\psi', a_i)$ in DNF
3: $\psi \leftarrow \varnothing$
4: **for** each disjunct $C \in \psi'$ **do**
5: $\quad \mathcal{D} \leftarrow \top$
6: $\quad$ **for** each literal $F_j(\vec{x_j})$ in $C$ **do**
7: $\quad\quad$ **Construct:** $F_\vdash^+, F_\vdash^-$
8: $\quad\quad F'_{a_i,j}(\vec{x}) \leftarrow \text{Regr}_\vdash(F_j(\vec{x}), a_i)$
9: $\quad\quad \mathcal{D} \leftarrow \mathcal{D} \wedge F'_{a_i,j}(\vec{x})$
10: $\quad\quad \mathcal{D} \leftarrow \text{ConvertToDNF}(\mathcal{D})$
11: $\quad$ **end for**
12: $\quad \psi \leftarrow \psi \vee \mathcal{D}$
13: **end for**
14: **return** $\psi$

---

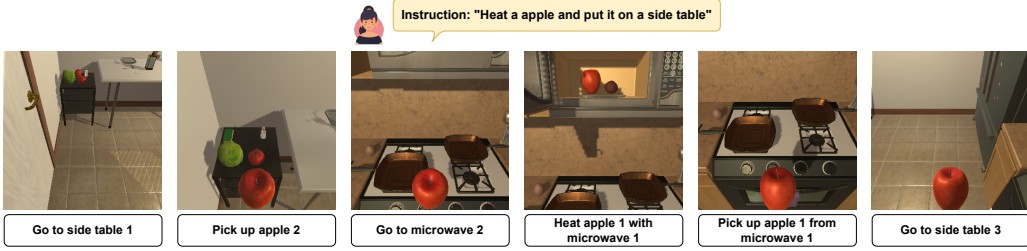

Figure 7: An Example of a Visual ALFWorld Task.

### 2.1.2 REGRESSION OF FORMULAS

Full first-order formulas can now be regressed using Algorithm 1: the formula is rewritten in Disjunctive Normal Form (DNF), the predicates in each disjunct are regressed following the process in Section 2.1.1, and the results are incrementally combined in DNF. The general regression process in the NL-PDDL framework is summarized in Appendix D.

## 3 EXPERIMENTS

We pose the following research questions:

RQ1: **Overall Planning Performance.** How does NL-PDDL[4] perform in terms of success rate and computational cost across modalities (e.g., text, images), compared to baselines?

RQ2: **Reasoning over Language Misalignment.** How well does NL-PDDL handle planning tasks where NL descriptions of the action specifications and the goal are misaligned?

RQ3: **Effect of Task Complexity.** How does NL-PDDL compare to baselines as task complexity increases, measured by optimal plan length and the number of constraints in the goal?

### 3.1 EXPERIMENTAL SETUP

**Open-World Tasks.** We conduct open-world planning experiments on *ALFWorld Text* (Shridhar et al., 2021) and *ALFWorld Vision* (cf. Figure 7) to evaluate performance across modalities. Agents must operate under partial observability and incomplete task-relevant knowledge to complete common household tasks in virtual home environments. It supports a rich space of interactions, requiring the agents to reason about object relations and action affordances without hardcoded annotations. Following prior work (Yang et al., 2024; Shridhar et al., 2021; Yao et al., 2023b; Wang et al., 2023), we evaluate NL-PDDL on 135 out-of-distribution tasks with 50 movement budget (cf. Appendix G).

---

[4]https://github.com/D3Mlab/NL-PDDL

Table 1: **(RQ1)** Performance of different methods in two **open-world** datasets **without misalignment** between goal and model.[†]Reported performance in the original work.

| | | | # Tok. | # Expert Trajectory | Fine-Tuning | SR (%) |
|---|---|---|---|---|---|---|
| **Datasets** | **Methods** | | | | | |
| | | **Open-World Without Misalignment** | | | | |
| **ALFWorld Text** | Direct LLM-based Planner | GPT-4o | 1,358,934 | 0 | ✗ | 21% |
| | | Gemini-2.0-Flash | 1,176,406 | 0 | ✗ | 16% |
| | | LLaMA-3.1 | 1,193,640 | 0 | ✗ | 19% |
| | Reflective LLM-based Planner | ReAct (w examples) | 5,507,266 | 0 | ✗ | 81% |
| | | ReAct (w model) | 4,242,560 | 0 | ✗ | 34% |
| | | Reflexion-3[†] | NA | 0 | ✗ | 83% |
| | | Reflexion-10[†] | NA | 0 | ✗ | 91% |
| | | DEPS[†] | NA | 0 | ✗ | 76% |
| | Fine-tuned LM | BUTLER | NA | 100,000 | ✓ | 26% |
| | **Ours** | **NL-PDDL** | 443,407 | 0 | ✗ | **94%** |
| **ALFWorld Vision** | Direct VLM-based Planner | GPT-4o | 1,823,539 | 0 | ✗ | 8% |
| | | Gemini-2.0-Flash | 1,440,025 | 0 | ✗ | 2% |
| | | LLaMA-3.1 | 1,505,371 | 0 | ✗ | 2% |
| | Fine-tuned VLM | LLaMA-Adapter[†] | NA | 170,000 | ✓ | 13% |
| | | InstructBLIP[†] | NA | 170,000 | ✓ | 22% |
| | | EMMA-3[†] | NA | 15,237 | ✓ | 37% |
| | | EMMA-10[†] | NA | 15,237 | ✓ | 82% |
| | **Ours** | **NL-PDDL** | 679,148 | 0 | ✗ | **84%** |

Table 2: **(RQ1)** Performance comparison across **closed-world** Blocksworld variants **without misalignment**. Direct LLM-based planners fail to generate valid plans for Mystery and Randomized Blocksworld. NL-PDDL demonstrates consistent and robust performance across all variants.

| | **Closed-World Without Misalignment** | | | | | |
|---|---|---|---|---|---|---|
| **Method** | **Blocksworld** | | **Mystery Blocksworld** | | **Randomized Blocksworld** | |
| | # Tok. | SR (%) | # Tok. | SR (%) | # Tok. | SR (%) |
| GPT-4o | 982,602 | 34% | 834,890 | 0% | 835,974 | 0% |
| Gemini-2.0 Flash | 928,024 | 18% | 834,060 | 1% | 835,824 | 0% |
| LLaMA 3.1 | 963,565 | 44% | 841,791 | 0% | 842,622 | 0% |
| Fast Downward | N/A | 100% | N/A | 100% | N/A | 100% |
| **NL-PDDL** | 0 | 70% | 0 | 70% | 0 | 70% |

We also extend both ALFWorld benchmarks by introducing misaligned language descriptions of the agent's action model and goals (cf. Appendix H).

**Closed-World Tasks.** *Blocksworld* is a widely used closed-world planning domain in which the task is to rearrange colored blocks, initially stacked or on a table, to achieve a given goal. We adopt a variant introduced in Valmeekam et al. (2022) by converting goal specifications into NL descriptions for NL-based planners. We include two additional variants to probe brittleness to surface form and lexical priors by replacing object and goal names in the NL descriptions with random English words (*Mystery Blocksworld*) and random strings (*Randomized Blocksworld*). Beyond these, we introduce *Misalignment Blocksworld*, where the NL descriptions of action model and goals are contextually meaningful but intentionally misaligned, to assess the commonsense entailment capabilities of planners (cf. Appendix H for details).

**Baselines and Evaluation Metrics.** We compare NL-PDDL with SOTA baselines in open- and closed-world domains. In the open-world settings (i.e., ALFWorld Text and Vision), we compare with (i) **Direct** LLM/VLM-based planners: GPT-4o (Hurst et al., 2024), Gemini-2.0 Flash (Team et al., 2023), and LLaMA-3.1 (Grattafiori et al., 2024), (ii) **Reflective** LLM-based planners: ReAct (Yao et al., 2023b), Reflexion (Shinn et al., 2023), and DEPS (Wang et al., 2023), and (iii) **Fine-tuned** LLM/VLMs: BUTLER (Shridhar et al., 2021), LLaMA-Adapter (Gao et al., 2023), InstructBLIP (Dai et al., 2023), and EMMA (Yang et al., 2024), all trained on ALFWorld expert demonsrations. In the closed-world setting, we compare against Direct LLM-based planners and the classical Fast Downward (Helmert, 2006) planner, which serves as an upper bound on planning performance. The detailed implementation of NL-PDDL on the ALFWorld and Blocksworld benchmarks is included in Appendix I. Methods are evaluated under a feasible maximum runtime. For evaluation, we consider two aspects: task *Success Rate* (SR), and associated cost, as captured by the *# Tokens* Usage (# Tok.) in LLM.

Table 3: **(RQ2)** Performance across settings **with misalignment** in open-world ALFWorld (Text/Vision) and closed-world Misalignment Blocksworld. Reported-results-only methods are omitted since prior works did not consider misalignment.

| | | Misaligned Benchmarks | | | | | |
|---|---|---|---|---|---|---|---|
| **Category** | **Method** | **ALFWorld Text** | | **ALFWorld Vision** | | **Blocksworld** | |
| | | # Tok. | SR (%) | # Tok. | SR (%) | # Tok. | SR (%) |
| Direct LLM | GPT-4o | 1,440,470 | 17% (↓5%) | 1,811,404 | 7% (↓1%) | 937,905 | 27% (↓7%) |
| | Gemini-2.0 Flash | 1,251,533 | 15% (↓1%) | 1,262,016 | 5% (↑3%) | 939,950 | 23% (↑5%) |
| | LLaMA-3.1 | 1,310,664 | 15% (↓4%) | 1,480,392 | 2% (-0%) | 950,490 | 42% (↓3%) |
| ReAct | w/ examples | 5,215,612 | 79% (↓12%) | N/A | N/A | N/A | N/A |
| | w/ model | 4,428,914 | 23% (↓11%) | N/A | N/A | N/A | N/A |
| Classical Planner | Fast Downward | N/A | N/A | N/A | N/A | 0 | 0% |
| **Ours** | **NL-PDDL** | 501,049 | **91%** (↓3%) | 745,365 | **80%** (↓4%) | 11,656 | **70%** |

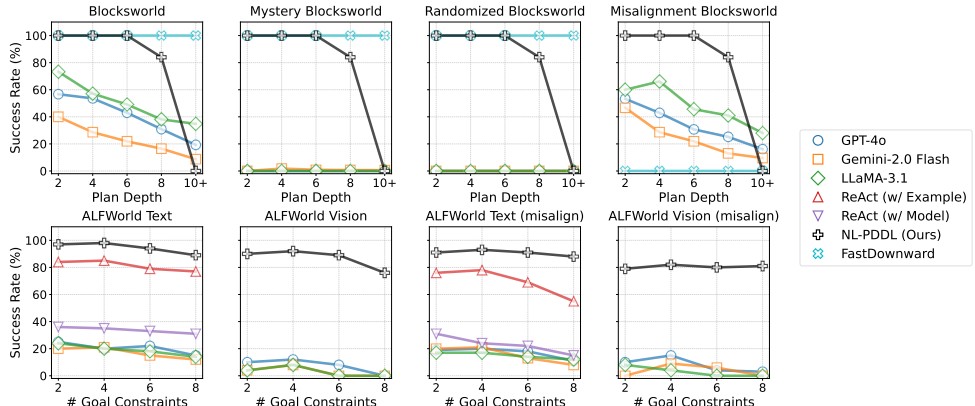

Figure 8: **(RQ3)** SR by goal complexity and optimal plan depth in ALFWorld and Blocksworld across planners

# 4 RESULTS AND DISCUSSION

**RQ1.** In open-world tasks (cf. Table 1), NL-PDDL achieves superior SR on both ALFWorld Text and Vision compared to all baselines. On ALFWorld Text, NL-PDDL achieves a 94% SR with only 443K tokens, whereas ReAct consumes over ten times more tokens while reaching a lower SR. Although Reflexion-10 achieves performance comparable to NL-PDDL, it depends on repeating ten trials of the same task, which is infeasible for many embodied AI applications. Impressively, NL-PDDL even outperforms all fine-tuned VLMs that rely on thousands of expert trajectories in visual ALFWorld. Our results highlight its generalizability across different modalities for open-world tasks. In closed-world tasks (cf. Table 2), NL-PDDL demonstrates robust performance with a consistent 70% SR at 0 tokens. In contrast, direct LLM-based planners entirely fail to solve the Mystery and Randomized variants, performing with less than 1% SR despite substantial token usage. Fast Downward, a closed-world planner, achieves a 100% success rate on the three Blocksworld domains. However, it is brittle under goal–action misalignment, as we demonstrate in RQ3.

**RQ2.** In Table 3, we evaluate the performance of different methods on open- and closed-world tasks with misalignment between the NL descriptions of action models and goals. NL-PDDL demonstrates clear robustness in the face of such misalignments. In open-world tasks, NL-PDDL maintains a 91% SR on ALFWorld Text and 80% on ALFWorld Vision. By contrast, ReAct, whether explicitly prompted to focus on action models or supported with few-shot examples, suffers notable degradation, underscoring its sensitivity to misalignment. The closed-world Misalignment Blocksworld variant highlights the gap: Fast Downward, though highly effective in purely symbolic and aligned settings, does not perform commonsense entailment and fails under goal—action schema misalignment; in contrast, NL-PDDL maintains performance comparable to other Blocksworld variants, evidencing robust misalignment handling.

**RQ3.** We evaluate planner performance relative to optimal plan depth. NL-PDDL achieves perfect success up to depth 6, with performance declining to 84% at depth 8 and fails beyond depth 10 under the runtime limit. By contrast, LLM-based planners exhibit consistent deterioration across all depths. NL-PDDL's decline stems from scalability limitations under the predefined maximum runtime, while LLM-based planners fail due to their inability to reason over an extended horizon. The Fast Downward planner maintains perfect performance when the action model and goals are fully aligned, but fails on the Misalignment Blocksworld where commonsense entailment reasoning is required. In open-world tasks where optimal depth is inconsistent, we measure NL-PDDL's performance against goal complexity. NL-PDDL remains stable, showing only a 3% average drop across ALFWorld Text and Vision, while LLM/VLM-based planners degrade much more sharply, underscoring their sensitivity to increasing goal complexity. Together, these results demonstrate the robustness of NL-PDDL with respect to goal complexity across open- and closed-world tasks.

## 5    RELATED WORK

**Eliciting Stronger Reasoning from LLMs:** As LLMs scale, they remain prone to hallucinations and logical fallacies  (Tonmoy et al., 2024; Zhang et al., 2025) resulting in new prompting strategies to mitigate these concerns. Step-by-step methods (Wei et al., 2022; Kojima et al., 2022; Yao et al., 2023a) encourage incremental reasoning by breaking problems into sub-steps or branching search trajectories. Reflective methods such as ReAct (Yao et al., 2023b) and Reflexion (Shinn et al., 2023) add self-reflection, critiquing, and revision. Yet these methods often fail on tasks with several logical constraints or long-horizon dependencies (Valmeekam et al., 2023b; 2022; Song et al., 2025); are sensitive to prompt design and few-shot examples, iteration dependent (Pan et al., 2023); and remain unverifiable since the LLM acts as a black box without guarantees of soundness (Shanahan, 2024).

**Planning with LLMs and Symbolic Planners:** A separate body of work integrates LLMs with classical planners. One approach treats LLMs as planners that generate action sequences from symbolic task descriptions (Silver et al., 2022; Song et al., 2023; Silver et al., 2024). Another uses LLMs as *translators*, converting NL problems and domains into structured formats for symbolic solvers (Guan et al., 2023; Kambhampati et al., 2024; Oswald et al., 2024; Yang et al., 2023). Both approaches face limitations: LLM-as-planner approaches are unreliable due to reasoning errors and hallucinations, while translator-based methods often introduce faulty predicates, actions, or logical mappings of NL descriptions that cause downstream planning failures. Translation also imposes an expressivity bottleneck, restricting applicability to problems encodable in symbolic form. Finally, most assume closed-world settings, making them inappropriate for open-world planning.

**Our Point of Departure:** We introduce a novel NL-PDDL hybrid natural language variation of PDDL to facilitate the use of powerful commonsense (V)LLM reasoning during planning, and blend it with symbolic *lifted* regression over NL-PDDL to address open-world, long-horizon planning.

We further discuss NL-to-PDDL translation approaches and clarify their distinctions from NL-PDDL in Appendix J.

## 6    CONCLUSION

We proposed NL-PDDL, a framework for open-world planning that extends symbolic PDDL to support NL, improving accessibility for non-experts. NL-PDDL facilitates the integration of LLM commonsense knowledge into open-world regression planning to address goal–action misalignment, while retaining soundness and verifiability. Empirically, we showed that lifted regression planning in NL-PDDL achieves higher plan-success rates in contrast to existing strong baselines, remains robust as plan horizons increase, and generalizes well across both text and vision modalities.

## ACKNOWLEDGMENTS

This work was supported by LG Electronics, Toronto AI Lab Grant Ref No. 2024-0565 and by the Institute of Information & Communications Technology Planning & Evaluation (IITP) grant funded by the Korean Government (MSIT) (No. RS-2024-00457882, National AI Research Lab Project).

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

## A    ETHICS, REPRODUCIBILITY, LLM USAGE STATEMENT

### A.1    ETHICS STATEMENT

We adhere to the ICLR Code of Ethics. This work does not involve human or animal subjects, and no identifiable personal information was used in developing this paper. Therefore, there are no privacy or security concerns. We are committed to maintaining integrity and transparency throughout the research process.

## A.2 Reproducibility Statement

We have made every effort to ensure that the methodology presented in this work is reproducible. All code used in this study has been released in an anonymous repository to support reproducibility and verifiability. Full descriptions of the methods and experimental setup are provided in both the main text and the appendix to facilitate transparent evaluation and replication of our results.

## A.3 The Use of Large Language Models (LLMs)

Large Language Models (LLMs) were used to aid or polish the writing of this paper. Specifically, we employed LLMs to improve the readability and clarity of the manuscript.

## B Fundamentals of Regression Planning in PDDL

In this section, we review PDDL and its representation of planning problems, then introduce the first-order regression approach to planning. The corresponding operations for our proposed NL-PDDL are presented in Section 2.1 of the main paper.

### B.1 PDDL Planning Representation.

A planning problem is defined as the tuple $\mathcal{P} = \langle \mathcal{F}, \mathcal{O}, \mathcal{A}, s_0, G \rangle$, where:

- $\mathcal{F}$ is the set of first-order predicates that describe properties of objects or relations between them, whose truth values may change as actions are applied, such as `boiled(x)`, `can_toast(x, y)`, etc.

- $\mathcal{O}$ is the set of objects in the domain, which serve as the constants over which predicates and actions are instantiated, (e.g., *potato*, *plate*, etc.).

- $\mathcal{A}$ is the set of actions $a_i(\vec{y})$, i.e., parameterized first-order operators. Each action schema is defined by:

  - *Preconditions:* Denoted by $a_i(\vec{y_i}).\texttt{pre} \subset \mathcal{F}$ is the set of predicates that must hold for the action to be applicable.
  - *Add effects:* Denoted by $a_i(\vec{y_i}).\texttt{add} \subset \mathcal{F}$ is the set of predicates that become true once the action is executed.
  - *Delete effects:* Denoted by $a_i(\vec{y_i}).\texttt{del}$ is the set of predicates that are no longer true after the action is executed.

  For instance, the action `pickup(?o)` has the preconditions `hand_empty` and `near(?o)`, the add effect `possess(?o)`, and the delete effect `hand_empty`.

- $s_0$ is the initial state, given as a set of ground predicates that hold true before any action is executed.

- $G$ is the goal condition, specified as a set of predicates that must be satisfied in a terminal state.

In classical planning, problems are often specified in the *Planning Domain Definition Language (PDDL)*, which is divided into two parts: the *domain* and the *problem instance*. The domain defines the agent's *action model* together with the set of predicates $\mathcal{F}$ and the set of action schemas $\mathcal{A}$. The *problem instance* specifies a finite object set $\mathcal{O}$, an initial state $s_0$ and $G$ given as a set of ground predicates.

The objective of solving a planning problem is to find a *conditional plan*, i.e., a finite sequence of actions $\pi = \langle a^{(1)}, \ldots, a^{(n)} \rangle$, such that, starting from $s_0$, executing each action in $\pi$ sequentially results in a state $s_n$ that satisfies the goal $G$. Here, the superscript $(i)$ in $a^{(i)}$ indicates that $a$ is the action executed at the $i$-th step of the plan, i.e., it represents the horizon index of the action within the sequence.

---

**Example.** We represent the planning problem as

$$\mathcal{P} = \langle \mathcal{F}, \ \mathcal{O}, \ \mathcal{A}, \ s_0, \ G \rangle.$$

**predicates $\mathcal{F}$.**

$$\mathcal{F} = \{\texttt{hand\_empty, near}(x), \texttt{possess}(x), \texttt{on}(x,y), \texttt{isHot}(x) \ \texttt{can\_bake}(x,y),$$
$$\texttt{full\_of\_water}(y), \texttt{baked}(x), \ \texttt{boiled}(x)\}.$$

**Objects $\mathcal{O}$.**

$$\mathcal{O} = \{ \textit{bread, plate, toaster, pot} \}.$$

**Actions $\mathcal{A}$.**

**pickup(?o) :**

        pre: $\{\texttt{hand\_empty , near(?o)}\}$
        add: $\{\texttt{possess(?o)}\}$
        del: $\{\texttt{hand\_empty}\}$

**puton(?o, ?r):**

        pre: $\{\texttt{possess(?o), near(?r)}\}$
        add: $\{\texttt{hand\_empty, on(?o,?r)}\}$
        del: $\{\texttt{possess(?o)}\}$

**toast(?o, ?r):**

        pre: $\{ \texttt{can\_toast(?o,?r), possess(?o)} \}$
        add: $\{\texttt{isHot(?o)}\}$
        del: $\varnothing$

**Initial state $s_0$.**

$s_0 = \{\texttt{hand\_empty, near(potato), near(plate), near(oven), near(pot)},$
    $\texttt{boiling\_device(pot), can\_toast(bread,toaster), full\_of\_water(pot)}\}.$

**Goal $G$.**

$$G = \{ \texttt{heated(bread), on(bread,plate)} \}.$$

---

## B.2 First Order Regression

*First-order Regression* addresses open-world planning by lifting object representations into variables and reasoning backward from goals. First-order regression allows the agent to generate lifted plans without requiring prior knowledge of the full object set $\mathcal{O}$ or a complete initial state $s_0$. In this approach, beginning from the goal, the objective is to iteratively regress the target formula through applicable actions, in order to compute the subgoals that must hold in earlier states. By repeating this process, the agent constructs a sequence of actions that achieves the original goal while remaining fully symbolic and lifted, enabling reasoning over unknown or partially observed objects.

In this section, we introduce the procedure for translating a PDDL problem into a first-order representation and explain how regression is performed using the Successor State Axioms (SSAs).

### B.2.1 PDDL to First-Order Logic Domain Theory

To axiomatize a PDDL domain theory, we begin by defining positive and negative effect axioms, which specify how predicates change as a result of actions.

- **Positive effect axioms**: state which actions $a \in \mathcal{A}$ can explicitly make each predicate $F(\vec{x})$ true.

- **Negative effect axioms** state which actions $a \in \mathcal{A}$ can explicitly make the predicate $F(\vec{x})$ false.

These axioms are essential for accurately regressing a goal through an action: to determine the conditions that must have held before the action, one needs to know exactly how each action modifies the state. To obtain these axioms, we make the following assumptions.

- No new variables outside of action parameter are introduced in preconditions and effects.
- No quantifiers are used in the action's preconditions and effects.
- All unquantified variables are implicitly universally quantified.

We assume positive and negative effect axioms can be specified by considering all of the ways in which each action can affect each predicate. Let $F(\vec{y_F})$ be a predicate in the current state, $\gamma^+_{F,a_i}(\vec{y}, s)$ is a first-order formula such that, if it holds in the current state $s$, then $F'(\vec{y_F})$ holds after executing $a_i(\vec{y})$. Similarly, let $\gamma^-_{F,a_i}(\vec{y}, s)$ be a first-order formula such that, if it holds in the current state $s$, then $\neg F'(\vec{y_F})$ holds after executing $a_i(\vec{y})$. We write $\gamma^+_{F,a_i}(\vec{y})$ and $\gamma^-_{F,a_i}(\vec{y})$ as shorthands for the state-dependent conditions $\gamma^+_{F,a_i}(\vec{y}, s)$ and $\gamma^-_{F,a_i}(\vec{y}, s)$ in the current state, respectively, and define the normal form of effect axioms as following:

$$\forall \vec{y} : \vec{T} \left[ \gamma^+_{F,a_i}(\vec{y}) \Rightarrow F'_{a_i}(\vec{y_F}) \right], \qquad \forall \vec{y} : \vec{T} \left[ \gamma^-_{F,a_i}(\vec{y}) \Rightarrow \neg F'_{a_i}(\vec{y_F}) \right]. \tag{7}$$

Here, $\vec{y} : \vec{T}$ indicates that each variable in $\vec{y}$ is assigned its type by a corresponding element in $\vec{T}$. Given a PDDL problem $\mathcal{P}$ with an action set $\mathcal{A}$, we can derive effect axioms in the aforementioned normal form, action by action. Consider an action $a(\vec{y}) \in \mathcal{A}$, and let $F(\vec{y_F})$ be a predicate in its positive effect set, i.e., $F(\vec{y_F}) \in a(\vec{y}).\texttt{add}$. The predicate parameters $\vec{y_F}$ correspond to a permutation of a subset of the action parameters $\vec{y}$. We use $\vec{y}_{\backslash F}$ to denote the remaining action parameters that do not appear in $\vec{y_F}$ which are existentially quantified. With this setup, we construct the following implication:

$$\text{for each } F(\vec{y_F}) \in a_i(\vec{y}).\texttt{add} : \ \forall \vec{y} : \vec{T}, a \ \underbrace{\left[ a = a_i(\vec{y}) \ \wedge \bigwedge_{\text{Pre}_j(\vec{y_j}) \in a_i(\vec{y}).\texttt{pre}} \text{Pre}_j(\vec{y_j}) \right]}_{\gamma^+_{F,a_i}(\vec{y})} \Rightarrow F'_{a_i}(\vec{y_F}).$$

$$\tag{8}$$

Similarly, for the predicates in the delete effects of the action, we have:

$$\text{for each } F(\vec{y_F}) \in a_i(\vec{y}).\texttt{del} : \ \forall \vec{y} : \vec{T}, a \ \underbrace{\left[ a = a_i(\vec{y}) \ \wedge \bigwedge_{\text{Pre}_j(\vec{y_j}) \in a_i(\vec{y}).\texttt{pre}} \text{Pre}_j(\vec{y_j}) \right]}_{\gamma^-_{F,a_i}(\vec{y})} \Rightarrow \neg F'_{a_i}(\vec{y_F}).$$

$$\tag{9}$$

We can combine multiple predicates with the same name via a disjunction. The detailed methodology is outlined in Appendix C.

---

**Positive effect axiom.** For $\texttt{isHot(?o)}$ with respect to the action $\texttt{toast(?o, ?r)}$:

$$\forall \texttt{?o:object}, \texttt{?r:object}, a \left( \underbrace{a = \texttt{toast(?o, ?r)} \ \wedge \ \texttt{can\_toast(?o, ?r)} \ \wedge \ \texttt{possess(?o)}}_{\gamma^+_{\texttt{isHot, toast}}(\texttt{?o,?r})} \right.$$

$$\left. \Rightarrow \texttt{isHot'(?o)} \right).$$

**Negative effect axiom.** For $\texttt{possess(?o)}$ with respect to the action $\texttt{puton(?o, ?r)}$:

$$\forall \texttt{?o:object}, \texttt{?r:receptacle}, a \left( \underbrace{a = \texttt{puton(?o, ?r)} \ \wedge \ \texttt{near(?r)} \ \wedge \ \texttt{possess(?o)}}_{\gamma^-_{\texttt{possess, puton}}(\texttt{?o,?r})} \right.$$

$$\left. \Rightarrow \neg\texttt{possess'(?o)} \right).$$

---

**Succesor State Axioms**    When a PDDL problem is transformed into the above normal form, Reiter (1991) showed that, under the assumptions of the *Unique Names Axioms* (Reiter, 1980) and the *Explanation Closure Axioms* (Reiter, 1991), we can construct *Successor State Axioms* (SSAs) that capture how a predicate may change or persist as the agent interacts with the environment. Let $F'_{a_i}(\vec{y_F})$ denote the value of a predicate $F(\vec{y_F})$ in the next state after executing action $a_i(\vec{y})$. The SSA for $F(\vec{y_F})$ is defined as:

$$\forall \vec{y} : \vec{T} \; [F'_{a_i}(\vec{y_F}) \equiv \gamma^+_{F,a_i}(\vec{y}) \; \vee \; \left(F(\vec{y_F}) \wedge \neg\gamma^-_{F,a_i}(\vec{y})\right)]. \tag{10}$$

Intuitively, SSAs state that a predicate $F'(\vec{y_F})$ in the next state can be true either because it is made true by an action $a_i(\vec{y})$, as specified by $\gamma^+_{F,a_i}(\vec{y})$, or because it was already true in the previous state, $F(\vec{y_F})$, and the action does not make it false, i.e., $\neg\gamma^-_{F,a_i}(\vec{y})$.

$$
\begin{array}{l}
\forall\,\texttt{?o:object, ?r:receptacle, } a \\[4pt]
\Big[\texttt{possess'(?o)} \equiv \underbrace{\left(a = \texttt{pickup(?o)} \wedge \texttt{hand\_empty} \wedge \texttt{near(?o)}\right)}_{\gamma^+_{\texttt{possess, pickup}}(\texttt{?o})} \\[18pt]
\qquad \vee \; \Big(\texttt{possess(?o)} \wedge \neg \underbrace{\left(a = \texttt{puton(?o,?r)} \wedge \texttt{near(?r)}\right)}_{\gamma^-_{\texttt{possess, puton}}(\texttt{?o,?r})}\Big)\Big].
\end{array}
$$

**First-Order Regression**    Let $\psi'$ denote a first-order state description that holds *after* executing an action $a_i$. The regression operator $Regr(\psi', a_i)$ "backprojects" $\psi'$ to compute a logical formula $\psi$ that must hold *before* the execution of $a_i$. Fortunately, SSAs provide us a logically equivalent pre-action condition for each predicate $F'_{a_i}(\vec{x}) \in \psi'$ with respect to an action $a_i \in \mathcal{A}$. Regression of the entire formula $\psi'$ is performed by recursively replacing each post-action predicate $F'(\vec{x})$ with its corresponding precondition formula $F'_{a_i}(\vec{x})$, as defined by the appropriate SSA.

$$Regr(F'(\vec{x}), a_i) \equiv F'_{a_i}(\vec{x}) \tag{11}$$

$$Regr(\texttt{isHot(?o),toast(?o, ?r)}) \equiv \texttt{can\_toast(?o,?r)} \wedge \texttt{possess(?o)}$$

SSAs lay the foundation of lifted regression planning, since using SSAs, we are able to replace a predicate in a post-action state with the subgoals that are required to hold before the action is executed to achieve the predicate. We provide examples of this functionality in Section 2.1.

## C    COMBINING EFFECT AXIOMS

**Combining Effect Axioms(Appendix)**    For predicates that appear in multiple action effects, we need to combine them into a single effect axiom. For instance, we have two antecedent condition formulae $C_1(\vec{x}_1, \vec{y}_1)$ and $C_2(\vec{x}_2, \vec{y}_2)$ where:

$$\forall \vec{x}_1 : \vec{T_{x_1}}, \vec{y}_1 : \vec{T_{y_1}} \; [C_1(\vec{x}_1, \vec{y}_1) \Rightarrow F(\vec{x}_1)], \tag{12}$$

$$\forall \vec{x}_2 : \vec{T_{x_2}}, \vec{y}_2 : \vec{T_{y_2}} \; [C_2(\vec{x}_2, \vec{y}_2) \Rightarrow F(\vec{x}_2)]. \tag{13}$$

let $\theta = \{\vec{x}_2 \mapsto \vec{x}_1, y_2 \mapsto \vec{y}_1\}$ be the most general unifier (MGU) of $C_1$, and $C_2$, i.e., the substitution that unifies the variables of $C_2$ with those of the $C_1$ without introducing unnecessary restrictions. We then apply $\theta$ to $C_2(\vec{x}_2, \vec{y}_2)$, i.e., $\text{SUBST}(\theta, C_2(\vec{x}_2, \vec{y}_2)) = C_2(\vec{x}_1, \vec{y}_1)$, and form the following implication:

$$\forall \vec{x_1} : \vec{T_{x_1}}, \vec{y_1} : \vec{T_{y_1}} \; [C_1(\vec{x}_1, \vec{y}_1) \vee C_2(\vec{x}_1, \vec{y}_1)] \Rightarrow \gamma^+_{F,a}(\vec{x}_1, \vec{y}_1), \tag{14}$$

where $\text{SUBST}(\theta, p)$ denotes the formula obtained by applying substitution $\theta$ to $p$.

```
∀a : A, ?o : object, ?r : object [
  (a = bake(?o,?r) ∧ baking_device(?r) ∧ can_bake(?o,?r) ∧ possess(?o))

  ∨

  (a = boil(?o,?r) ∧ boiling_device(?r) ∧ full_of_water(?r) ∧ possess(?o))]

    ⇒ isHot'(?o)
```

# D  NL-REGRESSION ALGORITHM

Algorithm 2 presents the regression process in the NL-PDDL framework. Starting from $G$, we standardize actions and regress the current subgoal $\psi'$ through them. The result is simplified and if valid, the action is appended to the plan, forming $(\psi, \vec{a})$. This process is repeated to depth $h$, yielding the final plan $\Pi$ which consists of all such pairs $(\psi, \vec{a})$ where $\psi$ is logically equivalent to the original goal $G$.

---

**Algorithm 2** NL-REGRESSION ALGORITHM

---

1: **Input:** $T = \langle G, \mathcal{A}, h, \mathcal{F} \rangle$
2: **Output:** $\Pi \subseteq \{(\psi, \vec{a})\}$ with each $\psi$ in DNF
3: $\Phi = \{F'_{a_i}(\vec{x}) \mid F(\vec{x}) \in \mathcal{F}, a_i(\vec{y}) \in \mathcal{A}\}$.
4: $Frontier \leftarrow \{(G, [], 0)\}$
5: $\Pi \leftarrow \{(G, [], \varnothing)\}$
6: **while** $Frontier \neq \varnothing$ **do**
7:     Extract $(\psi', \vec{a}, d)$ from $Frontier$
8:     **if** $d \geqslant h$ **then**
9:         **continue**
10:     **end if**
11:     **for** each $a_i(\vec{y}) \in \mathcal{A}$ **do**
12:         $a_i(\vec{y}) \leftarrow \text{STANDARDIZE}(a_i(\vec{y}))$         //See Appendix F.1
13:         $\psi \leftarrow \text{REGRESSFORMULA}(\psi', a_i(\vec{y}), \mathcal{F})$
14:         **if** $\psi$ is evaluated to False **then**
15:             **continue**
16:         **end if**
17:         $\psi \leftarrow \text{SIMPLIFY}(\psi)$         *//See Appendix F.2*
18:         $\vec{a}' \leftarrow \vec{a} \parallel [a_i(\vec{y})]$
19:         $Frontier \leftarrow Frontier \cup \{(\psi, \vec{a}', d+1)\}$
20:         $\Pi \leftarrow \Pi \cup \{(\psi, \vec{a}')\}$
21:     **end for**
22: **end while**
23: **return** $\Pi$

---

# E  PROOF OF FORMULAS FOR NL REGRESSION

In this appendix, we provide formal proofs for the NL regression process in the NL-PDDL framework (Equations 3– 6 and Figures 5 and 6 in the main paper).

## E.1  REGRESSION OF POSITIVE NL PREDICATES

**LLM-derived and Domain Axioms**   Assume the NL entailment relationship $F(\vec{x}) \vdash_{\text{LLM}} P(\vec{x})$,

$$\forall x \; F(x) \Rightarrow P(x), \tag{1}$$

type entailment $t_1 \vdash_{\text{LLM}} t_2$,

$$\forall x \; t_1(x) \Rightarrow t_2(x), \tag{2}$$

and the typed quantifier interpretation:

$$\exists x : r \; Q(x) \Leftrightarrow (\exists x \; r(x) \wedge Q(x)), \tag{3a}$$

$$\forall x : r \; Q(x) \Leftrightarrow (\forall x \; r(x) \Rightarrow Q(x)). \tag{3b}$$

We note that goals are existentially quantified and follow the $\exists x : t_1 P(x)$ typing semantics, meaning that all typed variables in goals have conjunctive typing semantics. Below, $x$ is left as a conjunctively typed, shared free variable since it is shared among different subgoals and cannot be quantified standalone.

**Lemma 1** *For free variable $x$, if $t_1(x) \wedge F(x)$ holds then $t_2(x) \wedge P(x)$ must also hold.*

**Proof 1** *Assume:*
$$t_1(x) \wedge F(x), \tag{4}$$
*From Equations (2) and (4):*
$$t_2(x), \tag{5}$$
*by Modus Ponens.*

*From Equations (1) and (4):*
$$P(x), \tag{6}$$
*by Modus Ponens.*

*Combining Equations (5) and (6):*
$$t_2(x) \wedge P(x), \tag{7}$$
*by deduction.*

**Conclusion.** We can regress type quantified goal $P(x)$ as $F(x)$ following the rules of Figure 5 such that if $F(x)$ holds then $P(x)$ will hold without a type contradiction.

### E.2 REGRESSION OF NEGATIVE NL PREDICATES

**LLM-derived and Domain Axioms** Assume having the predicate entailment $P \vdash_{\text{LLM}} F$, i.e.,
$$\forall x \ P(x) \Rightarrow F(x), \tag{1}$$
which by contraposition, can be rewritten as
$$\forall x \ \neg F(x) \Rightarrow \neg P(x), \tag{1a}$$
and the type entailment $t_1 \vdash_{\text{LLM}} t_2$,
$$\forall x \ t_1(x) \Rightarrow t_2(x), \tag{2}$$
and the typed quantifier interpretation
$$\exists x : r \ Q(x) \Leftrightarrow (\exists x \ r(x) \wedge Q(x)), \tag{3a}$$
$$\forall x : r \ Q(x) \Leftrightarrow (\forall x \ r(x) \Rightarrow Q(x)). \tag{3b}$$

Again, we note that goals are existentially quantified and follow the $\exists x : t_1 \neg P(x)$ typing semantics.

**Lemma 2** *If $t_1(x) \wedge \neg F(x)$ holds then $t_2(x) \wedge \neg P(x)$ must hold without a type contradiction.*

**Proof 2** *Assume:*
$$t_1(x) \wedge \neg F(x). \tag{4}$$
*From Equations (2) and (4):*
$$t_2(x), \tag{5}$$
*by Modus Ponens.*

*From Equations (1a) and (4):*
$$\neg P(x), \tag{6}$$
*by Modus Ponens.*

*Combining Equations (5) and (6):*
$$t_2(x) \wedge \neg P(x), \tag{7}$$
*by deduction.*

**Conclusion.** We can regress goal $\neg P(x)$ to $\neg F(x)$ following the rules of Figure 6 such that if $\neg F(x)$ holds then $\neg P(x)$ will hold without a type contradiction.

### E.3 REGRESSION OF POSITIVE PREDICATE WITH MULTIPLE ENTAILMENTS

We omit types here, since they follow the same rules as the single entailment case.

**LLM-derived Axioms** Assume predicate entailments $F_i \vdash_{\text{LLM}} P$ for all $i = 1, \ldots, n$:

$$\forall x\ F_i(x) \Rightarrow P(x), \qquad i = 1, \ldots, n \tag{1}$$

**Lemma 3** *Under multiple entailments $F_i \vdash_{LLM} P$, for $i = 1, \ldots, n$, if $F_1(x) \vee \cdots \vee F_n(x)$ holds then $P(x)$ holds.*

**Proof 3** *From Equation (1):*

$$[\forall x\ F_1(x) \Rightarrow P(x)] \wedge \cdots \wedge [\forall x\ F_n(x) \Rightarrow P(x)]. \tag{2}$$

*Renaming under the universal quantifier:*

$$\forall x\ [F_1(x) \Rightarrow P(x)] \wedge \cdots \wedge [F_n(x) \Rightarrow P(x)]. \tag{3}$$

*Rewriting $\Rightarrow$ as $\vee$:*

$$\forall x\ [\neg F_1(x) \vee P(x)] \wedge \cdots \wedge [\neg F_n(x) \vee P(x)]. \tag{4}$$

*Applying the reverse distributive law:*

$$\forall x\ [\neg F_1(x) \wedge \cdots \wedge \neg F_n(x)] \vee P(x). \tag{5}$$

*Applying De Morgan's theorem:*

$$\forall x\ [\neg(F_1(x) \vee \cdots \vee F_n(x))] \vee P(x). \tag{6}$$

*Rewriting $\vee$ as $\Rightarrow$:*

$$\forall x\ (F_1(x) \vee \cdots \vee F_n(x)) \Rightarrow P(x). \tag{7}$$

**Conclusion.** Under multiple entailments $F_i \vdash_{\text{LLM}} P$, for $i = 1, \ldots, n$, we can regress $P(x)$ to:

$$F_1(x) \vee \cdots \vee F_n(x).$$

### E.4 REGRESSION OF NEGATIVE PREDICATE WITH MULTIPLE ENTAILMENTS

We omit types here, since they follow the same rules as the single entailment case.

**LLM-derived Axioms** Predicate entailment $P \vdash_{\text{LLM}} F_i$ for all $i = 1, \ldots, n$:

$$\forall x\ P(x) \Rightarrow F_i(x), \qquad i = 1, \ldots, n \tag{1}$$

Contrapositive:

$$\forall x\ \neg F_i(x) \Rightarrow \neg P(x), \qquad i = 1, \ldots, n \tag{1a}$$

**Lemma 4** *Under multiple entailments $P \vdash_{LLM} F_i$, for $i = 1, \ldots, n$, if $\neg F_1(x) \vee \cdots \vee \neg F_n(x)$ holds then $\neg P(x)$ holds.*

**Proof 4** *From Equation (1a):*

$$[\forall x\ \neg F_1(x) \Rightarrow \neg P(x)] \wedge \cdots \wedge [\forall x\ \neg F_n(x) \Rightarrow \neg P(x)]. \tag{2}$$

*Renaming under the universal quantifier:*

$$\forall x\ [\neg F_1(x) \Rightarrow \neg P(x)] \wedge \cdots \wedge [\neg F_n(x) \Rightarrow \neg P(x)]. \tag{3}$$

*Rewriting ⇒ as ∨:*

$$\forall x \, [F_1(x) \lor P(x)] \land \cdots \land [F_n(x) \lor P(x)]. \tag{4}$$

*Applying the reverse distributive law:*

$$\forall x \, [F_1(x) \land \cdots \land F_n(x)] \lor P(x). \tag{5}$$

*Applying De Morgan's theorem:*

$$\forall x \, [\neg(\neg F_1(x) \lor \cdots \lor \neg F_n(x))] \lor P(x). \tag{6}$$

*Rewriting ∨ as ⇒:*

$$\forall x \, (\neg F_1(x) \lor \cdots \lor \neg F_n(x)) \Rightarrow \neg P(x). \tag{7}$$

**Conclusion.** Under multiple entailments $P \vdash_{\text{LLM}} F_i$, for $i = 1, \ldots, n$, we can regress $\neg P(x)$ to:

$$\neg F_1(x) \lor \cdots \lor \neg F_n(x).$$

# F  OPERATIONS IN FIRST-ORDER REGRESSION

## F.1  STANDARDIZATION

Let $\varphi$ be a first-order formula. The *standardization* of $\varphi$, denoted $\text{STANDARDIZE}(\varphi)$, is obtained by renaming the bound variables in $\varphi$ with fresh variables so that no two distinct quantifiers in $\varphi$ bind the same variable symbol.

Formally, if $x_1, \ldots, x_k$ are the bound variables in $\varphi$, then

$$\text{STANDARDIZE}(\varphi) = \varphi[\rho(x_1), \ldots, \rho(x_k)],$$

where $\rho$ is a bijective renaming function mapping each $x_i$ to a fresh variable $x_i'$ such that $x_i' \notin \text{Var}(\varphi)$ for all $i$. Here, $\text{Var}(\varphi)$ denotes the set of variables occurring in $\varphi$.

## F.2  SIMPLIFICATIONS

Regression typically generates a large, expanded DNF formula. To keep the resulting plan compact and interpretable, we apply a set of simplification rules that maintain logical equivalence while eliminating redundancy. These simplifications enable the planner to express subgoals clearly and prevent repeated formulas from arising during regression.

### F.2.1  EQUALITY-BASED QUANTIFIER ELIMINATION

We eliminate quantified variables that are equal to an object by replacing them with the object, i.e.,

$$\exists x : T, [x = x^* \land F(x)] \equiv F(x^*)$$

where $x^*$ is an object. This transformation preserves logical equivalence while eliminating redundant quantification. It is often applied during regression, where it is triggered by the need to unify variables between predicates and action parameters. This rule is especially valuable when the regressed formula includes equality constraints that enforce correspondences between variables.

### F.2.2  CONTRADICTION EVALUATION

Contradictions often arise within regressed formulas and can be simplified to $\perp$ (false). We consider two main cases:

- **Conjunctive Formulas.** A conjunction that contains mutually exclusive literals or unsatisfiable equality conditions simplifies to false. For example:

$$\begin{aligned} F(x) \land \neg F(x) &\rightarrow \perp, \\ x = y \land x \neq y &\rightarrow \perp. \end{aligned}$$

These situations typically occur when inconsistent conditions are introduced through regression or substitution.

**Disjunctive Formulas.** In disjunctions, the false literal $\perp$ can be eliminated since it does not affect the overall satisfiability:

$$\perp \vee F(x) \vee \ldots \quad \equiv \quad F(x) \vee \ldots$$

### F.2.3 No No-Op Assumption.

We assume that the domain does not contain *no-op actions*. A no-op action is an action that leaves the state unchanged, so we assume that every action makes a meaningful change to the state. Hence, any disjunct in the regressed formula that is logically identical to the original goal can be safely eliminated.

$$\text{Regr}(G(x)) \quad \equiv \quad \bigvee_i \Psi_i^{DNF} \vee G(x) \quad \equiv \quad \bigvee_i \Psi_i'^{DNF}$$

This guarantees that plans are constructed only from informative regressions.

### F.2.4 DNF Subsumption.

Let $\phi$ be a DNF formula:

$$\phi = \bigvee_{i=1}^{n} C_i,$$

in which each clause $C_i$ is a conjunction of literals. We say clause $C_i$ *subsumes* another clause $C_j$ if

$$C_i \Rightarrow C_j.$$

In this simplification technique, we eliminate any clause $C_j$ for which there exists another clause $C_i$ such that $C_i \Rightarrow C_j$. Formally, if

$$\exists i \neq j, \quad \text{such that} \quad C_i \Rightarrow C_j,$$

the formula

$$\phi = \bigvee_{i=1}^{n} C_i$$

is simplified to

$$\phi' = \bigvee_i C_i \quad \text{such that} \quad \forall j, k : (j \neq k) \Rightarrow \neg(C_j \Rightarrow C_k)$$

For example, with DNF subsumption, we have

$$[F_1(\vec{x_1}) \wedge F_2(\vec{x_2})] \wedge [F_1(\vec{x_1}) \wedge F_2(\vec{x_2}) \wedge F_3(\vec{x_3})] \equiv [F_1(\vec{x_1}) \wedge F_2(\vec{x_2})].$$

This simplification allows the planner to just maintain the most general subgoal formulas while preserving correctness.

### F.2.5 Duplicate Detection.

To prevent inefficiency during plan generation, a process called duplicate detection ensures the planner doesn't repeatedly visit the same subgoal. Exploring logically equivalent but distinct branches would lead to wasted computation and overly complex policies.

A regressed formula is marked as a duplicate if it is structurally identical to one already seen, ignoring differences in how variables are named or how the conjuncts are ordered.

To enable this duplicate detection, every conjunctive formula is standardized by performing the following steps:

- *Canonicalizing order*: Sorting all predicates and their arguments into a fixed, canonical sequence.
- *Consistent variable mapping*: Assigning variables consistently across the formula (e.g., uniformly replacing $x, y, z$ with $v_1, v_2, v_3$ in corresponding positions).
- *Flattening*: Removing nested conjunctions where possible.

### F.2.6 DOMAIN AXIOMS.

We enforce domain-specific integrity constraints for the *Blocksworld* domain to ensure all regressed formulas remain physically consistent.

**Single-Object Holding.** The agent cannot hold two distinct blocks simultaneously:

$$\forall b_1, b_2, \ b_1 \neq b_2 \rightarrow \neg(\text{Holding}(b_1) \land \text{Holding}(b_2)).$$

**Hand Consistency.** The agent's hand cannot be both holding a block and empty:

$$\forall b, \ \neg(\text{Holding}(b) \land \text{HandEmpty}).$$

Any clause violating these axioms during regression is simplified to $\perp$ and removed from the DNF.

## G ALFWORLD DETAILS

### G.1 ALFWORLD TASKS

ALFWorld simulates a typical household environment and focuses on daily embodied AI tasks. Table 4 lists the supported task types along with their corresponding goal templates.

Table 4: Task-types and templated goal descriptions in ALFWorld.

| Task-type | Templates |
|---|---|
| Pick & Place | (a) put a {obj} in {recep}. |
| | (b) put some {obj} on {recep}. |
| Examine in Light | (a) look at {obj} under the {lamp}. |
| | (b) examine the {obj} with the {lamp}. |
| Clean & Place | (a) put a clean {obj} in {recep}. |
| | (b) clean some {obj} and put it in {recep}. |
| Heat & Place | (a) put a hot {obj} in {recep}. |
| | (b) heat some {obj} and put it in {recep}. |
| Cool & Place | (a) put a cool {obj} in {recep}. |
| | (b) cool some {obj} and put it in {recep}. |
| Pick Two & Place | (a) put two {obj} in {recep}. |
| | (b) find two {obj} and put them {recep}. |

- **Pick & Place** (e.g., "put a plate on the coffee table") — the agent must find an object of the desired type, pick it up, find the correct location to place it, and put it down there.

- **Examine in Light** (e.g., "examine a book under the lamp") — the agent must find an object of the desired type, locate and turn on a light source with the desired object in-hand.

- **Clean & Place** (e.g., "clean the knife and put in the drawer") — the agent must find an object of the desired type, pick it up, go to a sink or a basin, wash the object by turning on the faucet, then find the correct location to place it, and put it down there.

- **Heat & Place** (e.g., "heat a mug and put on the coffee table") — the agent must find an object of the desired type, pick it up, go to a microwave, heat the object by turning on the microwave, then find the correct location to place it, and put it down there.

- **Cool & Place** (e.g., "put a cool bottle on the countertop") — the agent must find an object of the desired type, pick it up, go to a fridge, put the object inside the fridge and cool it, then find the correct location to place it, and put it down there.

- **Pick Two & Place** (e.g., "put two pencils in the drawer") — the agent must find an object of the desired type, pick it up, find the correct location to place it, put it down there, then look for another object of the desired type, pick it up, return to the previous location, and put it down there with the other object.

## G.2 EXAMPLE TRAJECTORY

```
You are in the middle of a room. Looking quickly around you, you see a
drawer 15, a drawer 19,
a drawer 7, an armchair 1, ... and a drawer 10.
Your task is to: find two remotecontrol and put them in armchair.
> go to sidetable 2
You arrive at loc 34. On the sidetable 2, you see a remotecontrol 1.
> take remotecontrol 1 from sidetable 2
You pick up the remotecontrol 1 from the sidetable 2.
> go to armchair 1
You arrive at loc 1. On the armchair 1, you see nothing.
> put remotecontrol 1 in/on armchair 1
You put the remotecontrol 1 in/on the armchair 1.
> go to sofa 1
You arrive at loc 2. On the sofa 1, you see a newspaper 1, a pillow 1, and
a remotecontrol 2.
> take remotecontrol 2 from sofa 1
You pick up the remotecontrol 2 from the sofa 1.
> go to armchair 1
You arrive at loc 1. On the armchair 1, you see a remotecontrol 1.
> put remotecontrol 2 in/on armchair 1
You won!
```

## H   ENTAILMENT DETAILS

### H.1   ENTAILMENT DESIGN

In this section we outline how we misaligned the action model and the goal for both AFLWorld and Blocksworld domain.

**ALFWorld Entailment Predicates**   Instead of using symbolic predicates such as, we define each action effect and precondition directly in NL and allow multiple entailed variants. Hard string matching is replaced with entailment sets as follows:

handEmpty : {"the agent's hand is empty", "the agent is not holding anything"}

$in(o, r)$ : {"$o$ is in $r$", "$o$ is inside $r$", "$o$ is stored in $r$"}

$hot(o)$ : {"$o$ is heated", "$o$ is hot", "$o$ is baked"}

$washed(o)$ : {"$o$ is washed", "$o$ is clean", "$o$ is cleaned"}

$cooled(o)$ : {"$o$ is chilled", "$o$ is cool", "$o$ is cooled"}

$holding(o)$ : {"the agent is holding $o$", "the agent possesses $o$", "the agent has $o$ in possessing"}

$on(r)$ : {"$r$ is turned on", "$r$ is on", "$r$ is switched on"}

**Blocksworld Entailment Predicates).**   We express each predicate directly in NL and allow multiple entailed variants rather than relying on canonical symbols. The entailment sets are:

handEmpty : {"the agent is not holding any objects", "the agent is not holding anything"}

$clear(b)$ : {"block $b$ has nothing on top of it", "no block is on $b$"}

$onTable(b)$ : {"block $b$ sits directly on the table", "$b$ is on the table"}

$on(b_1, b_2)$ : {"block $b_1$ is directly above block $b_2$", "$b_1$ on $b_2$"}

$holding(b)$ : {"the agent possesses block $b$", "the agent is holding $b$"}

### H.2   NATURAL LANGUAGE DESCRIPTION OF ACTION MODELS

We convert each action schema into a natural language description that specifies its preconditions and effects. These NL descriptions are provided to all LLM- and VLM-based models. To test

alignment, we intentionally modify the descriptions to use phrases that entail the goal but do not exactly match the goal string, thereby requiring explicit entailment reasoning. Below we provide examples of the NL descriptions of both the ALFWorld and Blocksworld action models.

**Natural Language ALFWorld Action Model** :

```
The pickup action requires the agent's hand to be empty and results in the
agent holding the target object. The put action requires the agent to
already be holding an object and allows the object to be placed inside or
on top of another receptacle, after which the agent's hand becomes empty.
The heat action requires the agent to be holding an object and to be at a
heating device, producing the effect that the object becomes hot or baked.
The wash action requires the agent to be holding the object and to be at a
washing device, resulting in the object becoming washed. The chill action
requires the agent to be holding the object and to be at a chilling device,
resulting in the object becoming chilled. The light action requires the
agent to be holding the object and to be at a lighting device, producing
the effect that the object becomes illuminated. Finally, device state can
be toggled by turn on and turn off, which require the agent to be at the
device and result in it being switched on or off
```

**Natural Language Blocksworld Action Model** :

```
The pickup action requires the agent's hand to be empty and results in the
agent holding the target object. The put action requires the agent to
already be holding an object and allows the object to be placed inside or
on top of another receptacle, after which the agent's hand becomes empty.
The heat action requires the agent to be holding an object and to be at a
heating device, producing the effect that the object becomes hot or baked.
The wash action requires the agent to be holding the object and to be at a
washing device, resulting in the object becoming washed. The chill action
requires the agent to be holding the object and to be at a chilling device,
resulting in the object becoming chilled. The light action requires the
agent to be holding the object and to be at a lighting device, producing
the effect that the object becomes illuminated. Finally, device state can
be toggled by turn on and turn off, which require the agent to be at the
device and result in it being switched on or off
```

# I DESIGN DETAILS OF NL-PDDL ALFWORLD AND BLOCKSWORLD IMPLEMENTATION

## I.1 ALFWORLD

In addition to the core NL-PDDL planner, our ALFWorld agent consists of an VLM-based observation parser, a knowledge base (KB), and an LLM-based object grounder. The observation parser extracts a list of object names from either text or image input. We use Gemini-2.0-Flash (Team et al., 2023) for image observations parsing. After obtaining the object names, we use GPT-4o to instantiate these into NL-predicates and update the agent's KB. The KB is implemented using Py-Datalog (Carbonnelle, 2024). Whenever new knowledge is acquired, the agent queries subgoals generated by the NL-PDDL regression planner to determine a feasible plan to execute.

**Observation Parser** The observation parser is responsible for extracting candidate object names and their types from either textual descriptions or visual observations. In the ALFWorld Text setting, object names are directly provided in the environment description, so we simply parse this list to extract object references. In the ALFWorld Vision setting, the parser receives RGB frames from a simulated first-person camera. Here, we use Gemini-2.0-Flash (Team et al., 2023), a VLM capable of open-vocabulary object detection, to generate bounding boxes around objects of interest. Gemini is prompted with relevant object descriptions (e.g., "Find a cup", "Find something that can wash the

cup") derived from subgoals generated by the NL-PDDL planner, and it returns a list of detected objects with bounding boxes, labels, and confidence scores. The resulting object list serves as the input to the grounding stage but does not by itself produce predicates or symbolic facts.

**Object Grounder**    The object grounder bridges the gap between raw object names and the structured predicates required by the NL-PDDL planner. Given the list of parsed object tokens and their textual or visual context, we use GPT-4o to generate grounded natural language predicates that align with our lifted planning formalism. For example, given the input "mug is cleaned," the grounder infers the predicate `clean(mug_1)`. The grounder also infers action-relevant relations such as `canClean(sink_1, apple_1)` when provided with relevant context. This ensures that grounded facts are consistent with both the agent's environment and the symbolic schema expected by the planner.

**Knowledge Base**    The knowledge base (KB) stores grounded predicates and supports logical reasoning over them. We implement the KB using PyDatalog (Carbonnelle, 2024), a lightweight logic programming library for Python that supports declarative predicate logic, variable binding, and rule-based inference. During plan execution, the planner queries the KB to check whether a lifted subgoal can be satisfied with the current facts. If the KB entails a subgoal, we bind the variables in both the subgoal and the associated actions, and then execute the corresponding plan. This allows the planner to maintain consistency between abstract subgoals and grounded execution.

## I.2 BLOCKSWORLD

For the Blocksworld domain, we convert the original PDDL goal into a corresponding NL-PDDL formulation. We follow the methodology of Valmeekam et al. (2023a) to ensure consistency between symbolic and natural language representations. The lifted regression planner then produces a sequence of subgoals with a fixed horizon of 10, which are grounded and validated sequentially. Each subgoal and the agent's initial state are translated into PyDatalog and checked for logical consistency with the original PDDL specification. Once all subgoals are validated, we use the standard PDDL validator from Muise et al. (2022) to ensure correctness of the complete plan.

## I.3 LLM PROMPT FOR TYPE CONSISTENCY CHECK

```
ROLE:
    You are a helper agent in a common household setting.
    You are checking TYPE ENTAILMENT between two predicates' term types.
INSTRUCTION:
    1. If the candidate's term type set implies or matches the target's term
    type set for each corresponding term, answer YES.
    2. Consider synonyms and common-sense subtype relations (e.g.,
    'vegetable' entails 'food').
    3. If information is unknown, be conservative and answer NO unless it's
    very likely.
INPUT:
    - Target term types: {$predicates_in_action_model_type$}
    - Candidate term types: {$misaligned_goal_type$}
OUTPUT FORMAT:
    - Line 1: exactly YES or NO.
    - Line 2: Reason.
RESPONSE:
```

```
ROLE:
    You are a helper agent in a common household setting.
QUESTION:
    - if you know Predicate 2 "{$action_model_predicate$}" is true, can you
    imply Predicate 1 "{$misaligned_goal_predicate$}" is true?.
    - Respond with exactly "YES" if you think the statement is generally
    implied
    - Respond with "NO" if you think the statement is generally false
INPUT:
    - Predicate 1: "{$misaligned_goal_predicate$}"
    - Predicate 2: "{$action_model_predicat$e$}"
INSTRUCTION:
    1. Use the definition of the predicates to determine if Predicate 2
    implies Predicate 1.
    2. You know the following background to determine the specific
    information of the objects within Predicate 1 and Predicate 2: $goal
    predicates$
    3. When determining the response, consider the meaning of the Predicate
    1 and Predicate 2 with the type of the specific object each refers to in
    common contexts.
    4. Be creative and think outside the box. If there is just a typo
    between the two predicates, you should say Yes.
OUTPUT FORMAT:
    - Line 1: exactly YES or NO.
    - Line 2: Reason.
RESPONSE:
```

# J  WORKS ON TRANSLATING NL TO PDDL

PDDL has long been established as the primary standard for defining planning domains and problems in AI Haslum et al. (2019). However, authoring accurate PDDL domains and problem specifications is a resource-intensive process that requires human expertise. To facilitate this process, a growing body of work has explored translating NL descriptions of planning domains and problems into PDDL.

Early efforts include NLtoPDDL Miglani & Yorke-Smith (2020), a pipeline that leverages readily available NL data by combining pre-trained contextual embeddings with Deep Reinforcement Learning (DRL) techniques previously used to extract structured plans from NL. Another example is FPTCP Huo et al. (2020), which constructs a ternary template of NL sentences to extract actions and their associated objects in a human-in-the-loop framework.

With the emergence of large language models (LLMs), their strong language understanding and commonsense reasoning abilities have been used to further facilitate PDDL construction from NL. For instance, Xie et al. (2023) uses the in-context learning capabilities of LLMs to translate NL domains and problems into PDDL via few-shot examples, demonstrating that using LLMs solely for translation and delegating the planning step to a PDDL solver yields superior results compared to direct LLM-based planning. Similarly, Smirnov et al. (2024) employs an LLM to generate PDDL plans, but also introduces LLM-based consistency checks and error-correction loops to improve plan quality. More recently, advanced approaches such as Ada Wong et al. (2023) have been proposed. Rather than translating a pre-existing instruction or domain manual into a single PDDL file, Ada interactively learns adaptive planning representations. In this framework, high-level action abstractions and low-level controllers are jointly adapted to a given domain of planning tasks, guided by NL inputs.

Although these seminal works significantly improve the usability of PDDL, they are all limited to translating NL inputs into PDDL and then relying on an existing PDDL planner—a process that can be error-prone due to the inherent limitations of LLMs and the auxiliary techniques involved. In contrast, NL-PDDL directly extends the PDDL framework to natively support NL specifications, thereby eliminating the need for NL-to-PDDL translation. Moreover, prior approaches primarily focus on generating and refining closed-world domains, whereas NL-PDDL supports open-world planning with incomplete domain knowledge.

## K    NUMBER OF UNDEFINED PREDICATES IN ALFWORLD

In practical scenarios, misalignment between an agent's action model and the underlying task goals is common. Consequently, it is essential for a symbolic planner to incorporate LLM-based reasoning when necessary—a capability that NL-PDDL is explicitly designed to provide.

As shown in Table 5, In the ALFWorld domain, a total of 583 unique predicates appear across all tasks, with an average of 52.26 predicates per task. Yet the action model defines only 22 predicates—approximately 3.7% of all predicates observed in our experiments. The remaining predicates arise dynamically from the user's instructions and the agent's interaction with the environment.

Because these predicates are not predefined, existing symbolic planning approaches that require a fixed, closed-world predicate set become infeasible. In contrast, NL-PDDL is the only formal framework that removes the need for a predefined predicate vocabulary by leveraging LLM-based entailment reasoning to interpret and ground predicates on demand.

Table 5: Predicate statistics for the ALFWorld domain.

| Statistic | Value |
|---|---|
| Total Number of Predicates | 583 |
| Predefined Predicates | 22 |
| Predicates per Task | 52.26 |

## L    ENTILAMENT CHECK

We conducted an additional ablation study (cf. Table 6) to evaluate the correctness of the LLM when performing predicate-level affordance reasoning over natural-language predicates. We randomly selected 20 predicates from ALFWorld and asked a human annotator to manually provide three sentences that should be entailed by each predicate and three sentences that should not be entailed. This yielded 120 entailment–non-entailment pairs in total. We then evaluated all pairs using the NL-PDDL entailment procedure to assess the model's precision and robustness in predicate-level reasoning.

Table 6: Classification statistics for the ALFWorld predicate evaluation.

| Metric | Number of Instances |
|---|---|
| True Positive | 60 |
| True Negative | 58 |
| False Positive | 0 |
| False Negative | 2 |
| Total Accuracy | 98.3 |

## M    LLM CALLS AND LATENCY

We randomly select 20 goals from *Misaligned BlockWorld* and *Misaligned ALFWorld* for each depth and report the mean and 95% confidence interval of the total number of LLM calls in Table 7. These totals include both local cache hits (i.e., we store all previously encountered LLM entailments in a global cache shared across goals) and actual API calls.

Table 7: LLM call and entailment statistics across domains.

| Domain | ALFWorld with Misalignment | Misalignment BlocksWorld |
|---|---|---|
| LLM API Call | $50.4 \pm 12.9$ | $73.2 \pm 23.4$ |
| Total Entailments Reasoning | $221.5 \pm 155.6$ | $1263 \pm 435.7$ |
| Avg Latency per Call | 1.8648 sec | 1.2443 sec |

