# OpenReview forum: "Natural Language PDDL (NL-PDDL) for Open-world Goal-oriented Commonsense Regression Planning in Embodied AI"
_ICLR.cc/2026/Conference — ICLR 2026 Poster_

### Official Review · Reviewer_DC4D · 2025-10-31

**Soundness:** 3
**Presentation:** 2
**Contribution:** 2
**Rating:** 4
**Confidence:** 3

**Summary:**

To address the open-world embodied AI tasks with partial observability and incomplete knowledge, this paper proposes NL-PDDL, which combines the strictness of symbolic planning with the semantic flexibility of LLMs. NL-PDDL first extends PDDL into a natural language form for representation, enabling LLM-based commonsense entailment reasoning, and finally employs lifted regression planning. The method maintains robustness under goal-action specification misalignment and generalizes across both textual and visual modalities.

**Strengths:**

1. Maintain the verifiability of symbolic planning.
2. Handle semantic misalignment through LLM.
3. Lifted representation avoids exhaustive grounding.
4. Works across textual and visual modalities

**Weaknesses:**

1. Limited novelty compared to prior work. I believe NL-PDDL lacks sufficient innovation. LLM-Regress [1] already proposed combining lifted regression with LLM for open-world planning. What advantages does NL-PDDL offer over this approach? From both the methodology and experimental results, NL-PDDL's advantages are not prominent. LLM-Regress achieves higher success rates and lower token consumption on ALFWorld.

2. Scalability and applicability limitations. NL-PDDL is not a general, scalable approach for open-world planning, as PDDL cannot fully model all problems in open-world settings. In addition, with increasing problem complexity, a sharp performance degradation of NL-PDDL (Figure 7) is observed. This instability may limit the method's practical applicability.

3. Inability to learn. NL-PDDL lacks learning capability, relying entirely on predefined domain specifications and LLM commonsense without the ability to improve from experience or adapt to new domains automatically.

[1] Liu X, Pesaranghader A, Li H, et al. Open-world planning via lifted regression with LLM-inferred affordances for embodied agents

**Questions:**

1. Given that NL-PDDL relies on natural language representations, could semantic ambiguity or variability in NL descriptions cause the LLM-based entailment reasoning to produce incorrect unifications? For instance, if the entailment check between goal predicates and action effects yields false positives or false negatives due to ambiguous phrasing, would this result in invalid plans or missed valid action sequences during lifted regression?

2. Could NL-PDDL be integrated as a verifiable planning component within broader LLM-based planning frameworks? Would this hybrid architecture improve both the correctness guarantees and the adaptability to novel scenarios beyond the predefined PDDL domain specifications?

---

> ### Author Response · Authors · 2025-11-21
> **In Response to the Official Review**
>
> We thank the reviewer for their constructive and insightful comments. We address each point in detail below.
> ### **W1 (Reviewer Comment)**
> > LLM-Regress already proposed combining lifted regression with LLM for open-world planning
> ### **Response**
> LLM-Regress only solves a subset of open-world problems with the following limitations:
>
> 1. It only handles incomplete object affordances.
> 2. The goal and action descriptions must share the same rigid symbolic representation.
> 3. User instructions are assumed to perfectly align with the agent’s model.
> 4. Observations are restricted to textual inputs only.
>
> NL-PDDL can handle open-world settings without these limitations and makes the following novel contributions to open-world planning that are not addressed in existing work:
>
> 1. Provides a theoretically sound planning framework that operates directly over natural-language statements.
> 2. Supports generic entailment reasoning using LLM-based commonsense, beyond simple affordance reasoning.
> 3. Handles misalignment between the goal specification and the action model.
> 4. Generates sound and verifiable plans under multiple modalities of observation.
>
> Our experiments include **misaligned domains** that cannot be handled by any symbolic planner. NL-PDDL also substantially outperforms all LLM/VLM baselines, including those fine-tuned on domain-specific data.
>
> ### **W2 (Reviewer Comment)**
> > NL-PDDL is not a general, scalable approach for open-world planning, as PDDL cannot fully model all problems in open-world settings.
> ### **Response**
> While NL-PDDL extends from symbolic PDDL ( the de facto standard for closed-world planning [1, 2]), it does not inherit many of its modeling limitations. We accomplish this by:
>
> 1. Using natural language to represent general facts and world descriptions rather than rigid symbolic representations.
> 2. Leveraging LLM-based commonsense reasoning to infer object properties and relations, compensating for incomplete domain knowledge.
>
> Our method is also more scalable than grounded PDDL because we:
>
> 1. Employ lifted abstractions to reason about properties without requiring concrete object instances.
> 2. Avoid combinatorial explosion in the observation and state space by focusing only on relevant objects and relations.
>
> These advantages allow NL-PDDL to address a much broader range of open-world planning problems in a more scalable manner than standard PDDL planners, as demonstrated by our results.
>
> ### **W3 (Reviewer Comment)**
> > In addition, with increasing problem complexity, a sharp performance degradation of NL-PDDL (Figure 7) is observed. This instability may limit the method's practical applicability.
> ### **Response**
> The Blocksworld domain is designed to highlight the insufficiency of LLM reasoning in long-horizon, closed-world problems. We observe that all LLM-based baselines struggle, whereas NL-PDDL consistently maintains high performance across the three Blocksworld domains (cf. Table 2 in the manuscript).
>
> A closed-world planner such as FastDownward performs optimally when the Blocksworld domain is perfectly aligned. However, its performance drops to 0% under action–goal misalignment, while NL-PDDL maintains strong performance (cf. Table 3).
>
> NL-PDDL is explicitly designed for planning in open-world environments, evaluated using ALFWorld, where it significantly outperforms existing baselines, as reported in Tables 1 and 3.
> ### **W4 (Reviewer Comment)**
> > NL-PDDL lacks learning capability, relying entirely on predefined domain specifications and LLM commonsense without the ability to improve from experience or adapt to new domains automatically.
> ### **Response**
> NL-PDDL focuses on complex causal reasoning in open-world settings and does not require any additional training or fine-tuning of LLMs/VLMs. This makes NL-PDDL both easier to use and more cost-effective.
>
> NL-PDDL can naturally be extended to a paradigm where the underlying foundation model is fine-tuned for stronger entailment reasoning using successful plan traces. However, in our experiments, existing models already provide sufficient entailment capabilities, as NL-PDDL outperforms all baselines that do rely on fine-tuning with domain-specific data. Further exploration of fine-tuning is outside the scope of this work.
>
>
> ### **References**
>
> [1] Gerevini, Alfonso Emilio. *An Introduction to the Planning Domain Definition Language (PDDL): Book Review.* Artificial Intelligence 280 (2020): 103221.
> [2] Haslum, Patrik, et al. *An Introduction to the Planning Domain Definition Language.* Vol. 13. Morgan & Claypool Publishers, 2019.

---

> ### Author Response · Authors · 2025-11-21
> **In Response to the Official Review**
>
> ### **Q1 (Reviewer Comment)**
> > Given that NL-PDDL relies on natural language representations, could semantic ambiguity or variability in NL descriptions cause the LLM-based entailment reasoning to produce incorrect unifications?
> ### **Response**
> NL-PDDL performs **entailment reasoning on an atomic level** over predicates rather than relying on the LLM to reason over complex logical constraints. As a result, its entailment reasoning is substantially more reliable than directly generating plans with an LLM.
>
> To further illustrate this, we increased the number of entailment variants for ALFWorld tasks to evaluate the robustness of LLM-based entailment. We conducted an additional ablation study (cf. Table below or Appendix L) to assess the correctness of predicate-level affordance reasoning under variations of natural-language predicate expressions.
>
> We randomly selected 20 predicates from ALFWorld and asked a human annotator to provide three sentences that *should* be entailed by each predicate and three that *should not* be entailed. We then evaluated all 120 entailment pairs using the NL-PDDL entailment procedure.
>
> | Metric          | Number of Instances |
> |-----------------|------------------|
> | True Positive   | 60               |
> | True Negative   | 58               |
> | False Positive  | 0                |
> | False Negative  | 2                |
> | Total Accuracy  | 98.3             |
>
>
> Across all 120 evaluated instances, the entailment procedure remains highly reliable even under varied linguistic formulations. The LLM correctly identifies 60 true positives and 58 true negatives, with zero false positives and only 2 false negatives, yielding an overall accuracy of **98.3%**. This shows the robustness of LLM entailment reasoning.
>
> We also want to emphasize that NL-PDDL enables **verifiable planning**, a prominent feature that existing LLM-based methods lack. NL-PDDL generates explicit reasoning traces, and in rare cases where LLM generates incorrect entailment, it can be easily identified.
>
> ### **Q2 (Reviewer Comment)**
> > Could NL-PDDL be integrated as a verifiable planning component within broader LLM-based planning frameworks? Would this hybrid architecture improve both the correctness guarantees and the adaptability to novel scenarios beyond the predefined PDDL domain specifications?
> ### **Response**
> We agree with the reviewer that the ability to verify an agent’s actions and plans is crucial in real-world applications like household robotics and autonomous driving [1,2,3]. NL-PDDL can be seamlessly integrated into any LLM-based (or VLM/VLA) framework as a verifiable planner, serving as a modular sub-routine that produces plans for downstream tasks.
>
> Another way to utilize NL-PDDL is to generate sound plan traces for LLM  (or VLM/VLA) fine-tuning, as a way to enhance their structured reasoning and planning capabilities.
>
> We view these as promising directions for extending the applicability of NL PDDL, but they fall outside the scope of the current work. We plan to explore related problems in future research. If the reviewer has a specific methodology of ‘broader LLM-based planning frameworks’, we would be happy to discuss in more detail.
>
> ### **References**
>
> [1] Alterovitz, Ron, Sven Koenig, and Maxim Likhachev. "Robot planning in the real world: Research challenges and opportunities." Ai Magazine 37.2 (2016): 76-84.
>
> [2] García, Sergio, et al. "Software variability in service robotics." Empirical Software Engineering 28.2 (2023): 24.
>
> [3] Brewitt, Cillian. "Interpretable and verifiable planning and prediction for autonomous vehicles." (2023).

---

### Official Review · Reviewer_qrc8 · 2025-10-31

**Soundness:** 3
**Presentation:** 3
**Contribution:** 2
**Rating:** 4
**Confidence:** 4

**Summary:**

Paper proposes NL-PDDL, a hybrid framework that integrates natural-language representations with symbolic regression planning to enable open-world, goal-oriented reasoning for embodied AI. The authors extend classical PDDL by allowing goals, predicates, and actions to be expressed in natural language, while preserving logical structure for verifiable planning. A lifted regression algorithm combines symbolic reasoning with LLM-based commonsense entailment, enabling the planner to infer semantically aligned actions and to reason under partial observability without exhaustive grounding. Experiments demonstrate higher plan-success rates compared with baselines. The work aims to bridge the interpretability of symbolic planners with the flexibility of language models for open-world embodied reasoning.

**Strengths:**

1.Paper conducts experiments on 3 benchmarks demonstrate the effectiveness of the proposed approach.

2.By leveraging lifted regression, NL-PDDL maintains scalability with respect to the number of objects, a property desirable for open-world and partially observable environments.

**Weaknesses:**

1.Unclear LLM-call efficiency and runtime impact
The regression algorithm requires repeated LLM entailment checks for each predicate and clause. Although the paper reports token counts, it omits time latency and number of LLM calls per plan. It could difficult to assess whether the approach is practical for deployment or real-time settings without this.

2.No clear bridge from symbolic plan to robot execution
While the framework grounds entities in images via a VLM, it stops short of showing how regressed NL-PDDL actions map onto executable robot actions. The granularity of generated subgoals may not align with the level of abstraction supported by downstream policies.

3.Incomplete treatment of entailment aggregation
The “aggregate all entailing predicates” step depends on the finite predicate set and LLM entailment checks, but how to prove the sufficiency of this set.

4.No real-world or hardware validation
All experiments are performed in simulated domains and lack of real robot experiments.

**Questions:**

1.Efficiency and scalability
How many LLM entailment calls are typically made per plan? What is the average planning latency compared to other planners?

2.Execution interface:
How do NL-PDDL actions interface with a robot's control stack or low-level skill library? Can the planner adapt its decomposition granularity based on the robot's available primitives?

3.Entailment completeness:
How is the predicate set constructed, and how large is it in your experiments?

---

> ### Author Response · Authors · 2025-11-21
> **In response to the official review (1/2)**
>
> We thank the reviewer for their constructive and insightful comments. We address each point in detail below.
>
> ### **W1, Q1 (Reviewer Comment)**
> > *1. Unclear LLM-call efficiency and runtime impact. The regression algorithm requires repeated LLM entailment checks for each predicate and clause. Although the paper reports token counts, it omits time latency and number of LLM calls per plan. It could difficult to assess whether the approach is practical for deployment or real-time settings without this.*
> >
> > *1. Efficiency and scalability. How many LLM entailment calls are typically made per plan? What is the average planning latency compared to other planners?*
>
> ---
>
> ### **Response (W1 Q1: Efficiency and scalability of LLM entailment calls)**
>
> We clarify that **NL-PDDL is an offline planning approach** [1,2], where the full set of lifted conditional plans is generated **prior to task execution**. Consequently, our method is far less sensitive to LLM latency than online planning approaches, which require replanning during execution time [3, 4].
>
> We report token usage because it directly reflects the amount of inferential work required from the LLM [5] and is the most relevant metric for us because **LLM entailment inference is done prior to task execution**. Also, the token count is stable across hardware and deployment settings, unlike wall-clock latency.
>
> For completeness, we add the measurement of both the average number of LLM calls and wall-clock LLM call latency for tasks with misalignment in Appendix M. However, please note that wall-clock latency can vary substantially with server load, network conditions, and other API-level factors.
>
> **Domain Comparison (Misaligned Settings)**
>
> | Domain                   | LLM API Call (avg ± sd) | Total Entailments Reasoning (avg ± sd) | Avg Latency |
> |--------------------------|---------------------------|------------------------------------------|-------------|
> | ALFWorld - Misaligned    | 50.4 ± 12.9               | 221.5 ± 155.6                            | 1.8648 sec  |
> | BlocksWorld - Misaligned | 73.2 ± 23.4               | 1263 ± 435.7                             | 1.2443 sec  |
>
> ---
>
> ### **W2, Q2 (Reviewer Comment)**
> > *2. No clear bridge from symbolic plan to robot execution. While the framework grounds entities in images via a VLM, it stops short of showing how regressed NL-PDDL actions map onto executable robot actions. The granularity of generated subgoals may not align with the level of abstraction supported by downstream policies.*
> >
> > *2. Execution interface: How do NL-PDDL actions interface with a robot's control stack or low-level skill library? Can the planner adapt its decomposition granularity based on the robot's available primitives?*
>
> ---
>
> ### **Response (W2 Q2: No clear bridge from symbolic plan to robot execution)**
>
> NL-PDDL is based on the model structure of standard PDDL (see Appendix B) and **can be directly translated to a standard symbolic PDDL** representation. A large body of work in planning and robotics already uses PDDL as the task-level specification that is mapped to lower-level robotic controllers [9, 10, 11]. NL-PDDL inherits this property: once translated to symbolic PDDL, the same established action-mapping mechanisms apply.
>
> We also emphasize that NL-PDDL is a **task-level planner**, and lower-level control mapping is outside the scope of this work. Additionally, using virtual environments (e.g., images) is standard in task-level embodied agent planning [6, 7], and mapping between task-level planning to lower-level robotics control is often studied as separate topics [8].
>
> ---

---

> > ### Author Response · Authors · 2025-11-21
> > **In response to the official review (2/2)**
> >
> > ### **W3, Q3 (Reviewer Comment)**
> > > *3. Incomplete treatment of entailment aggregation. The “aggregate all entailing predicates” step depends on the finite predicate set and LLM entailment checks, but how to prove the sufficiency of this set.*
> > >
> > > *3. Entailment completeness: How is the predicate set constructed, and how large is it in your experiments?*
> >
> > ---
> >
> > ### **Response (W3 Q3: The “aggregate all entailing predicates” step)**
> >
> > We want to clarify that NL-PDDL **does not assume a fixed and predefined predicate set** (a requirement for classical planners), making it suitable for open-world tasks. NL-PDDL predicates come from two sources:
> > 1. The agent’s action model, which specifies the high-level skills available to the agent
> > 2. The predicates appearing from user instructions
> >
> > In practical scenarios, misalignment between the agent’s action model and the goal is very possible (see table below or Appendix K); hence, it is crucial to have the ability to integrate LLM reasoning during the symbolic process, which NL-PDDL addresses.
> >
> > Also, note that:
> > 1. While the action model is fixed, goal predicates are extracted from the agent's goal (which are expressed in natural language) and are unbounded.
> > 2. We are the first to introduce this commonsense entailment procedure in a formal planning framework, and it is not feasible in any existing formal approaches.
> >
> > **Predicate Statistics (ALFWorld)**
> >
> > | Metric                     | Value  |
> > |---------------------------|--------|
> > | Total Number of Predicates | 583    |
> > | Predefined Predicates      | 22     |
> > | Predicates per Task        | 52.26  |
> >
> > As we can see in the ALFWorld domain, a total of 583 unique predicates appear across tasks, with an average of 52.26 predicates per task. Yet the action model contains only 22 predicates, which is just about 3.7% percent of all predicates that arise during our experiments. The remaining predicates come directly from the user’s instruction and from its interaction with the environment. These predicates are not predefined, which makes any existing symbolic planning approach infeasible. Our NL-PDDL is the only formal framework that eliminates the need for a fixed set of predefined predicates through the use of LLM-based entailment reasoning.
> >
> > ---
> >
> > ### **W4 (Reviewer Comment)**
> > > *4. No real-world or hardware validation.All experiments are performed in simulated domains and lack of real robot experiments.*
> >
> > ---
> >
> > ### **Response (W4: No real-world or hardware validation)**
> >
> > First, as we discussed in W2, NL-PDDL is a **task-level planning approach** that focuses on high-level decision making rather than lower-level hardware implementation. Please note that this paper builds on a rich line of work (papers, workshops, competitions) [6, 7] that simulates world interaction through Embodied AI simulators.
> >
> > While hardware-level deployment would be our ultimate long-term goal, this paper focuses on the specific problem of verifiable symbolic planning methodologies leveraging NL representations and LLM-supported open world commonsense reasoning.
> >
> > We also want to highlight that our contribution is mainly the NL-PDDL framework itself which introduces a sound open-world planning formulation enabled by LLM-based entailment checking with natural language predicates, which is not feasible with existing symbolic or LLM-based approaches.
> >
> > ---
> >
> > ### **References**
> >
> > [1] Ghallab, Malik, Dana Nau, and Paolo Traverso. *Automated planning and acting.* Cambridge University Press, 2016.
> > [2] LaValle, Steven M. *Planning algorithms.* Cambridge University Press, 2006.
> > [3] Yao, Shunyu, et al. *ReAct: Synergizing reasoning and acting in language models.* ICLR 2022.
> > [4] Brenner, Michael, and Bernhard Nebel. *Continual planning and acting in dynamic multiagent environments.* 2006.
> > [5] Sui, Yang, et al. *Stop overthinking: A survey on efficient reasoning for large language models.* TMLR (2025).
> > [6] Liu, Yang, et al. *Aligning cyber space with physical world: A comprehensive survey on embodied AI.* IEEE/ASME T-Mech (2025).
> > [7] Duan, Jiafei, et al. *A survey of embodied AI.* IEEE TETCI (2022).
> > [8] Garrett, Caelan Reed, et al. *Integrated task and motion planning.* ARCRAS (2021).
> > [9] Kaelbling, Leslie Pack, and Tomás Lozano-Pérez. *Hierarchical task and motion planning in the now.* ICRA 2011.
> > [10] Cashmore, Michael, et al. *ROSPlan: Planning in the Robot Operating System.* ICAPS 2015.
> > [11] Srivastava, Siddharth, et al. *Combined task and motion planning through an extensible planner-independent interface layer.* ICRA 2014.

---

### Official Review · Reviewer_SAUU · 2025-11-01

**Soundness:** 2
**Presentation:** 3
**Contribution:** 3
**Rating:** 6
**Confidence:** 2

**Summary:**

This paper proposes NL-PDDL, an extension of classical PDDL for open-world goal-oriented planning in embodied AI. NL-PDDL represents goals and actions through typed natural language constructs to integrate LLM commonsense reasoning during planning. The framework avoids exhaustive grounding and leverags entailment-based unification to tackle misalignment between natural language goal specifications and action models. Experimental evaluation is conducted on ALFWorld (text and vision variants) and Blocksworld, comparing against SOTA LLM, VLM, and symbolic planning baselines across modalities, goal complexities, and specification alignments.

**Strengths:**

1. The idea of extending PDDL into a natural-language variant and embedding LLM-based entailment checks into the planning loop is natural and intuitive. This combines sound symbolic reasoning with the flexibility of language-based commonsense, addressing issues where classical PDDL or direct LLM planners individually fail.

2. The results on well-known datasets show that this form of representation works relatively well for autoregressive models. If this paradigm is proven to be more suitable for training, it could become a general paradigm.

**Weaknesses:**

1. There are some very relevant works that should be cited and compared, e.g. Ada (Learning adaptive planning representations with natural language guidance), and other works that translate between natural language and PDDL, or have LLMs generate PDDL for planning purposes, such as:
a. NLtoPDDL: One-Shot Learning of PDDL Models from Natural Language Process Manuals
b. GPTPDDL (Translating Natural Language to Planning Goals with Large-Language Models),
c. Generating consistent PDDL domains with Large Language Models,
..... and many more

**Questions:**

N/A

---

> ### Author Response · Authors · 2025-11-21
> **In response to the official review**
>
> We thank the reviewer for their constructive and insightful comments. We address each point in detail below.
>
> **W1 (Reviewer Comment)**
> > *"There are some very relevant works that should be cited and compared, e.g. Ada (Learning adaptive planning representations with natural language guidance), and other works that translate between natural language and PDDL, or have LLMs generate PDDL for planning purposes, such as: a. NLtoPDDL: One-Shot Learning of PDDL Models from Natural Language Process Manuals b. GPTPDDL (Translating Natural Language to Planning Goals with Large-Language Models), c. Generating consistent PDDL domains with Large Language Models, ..... and many more*
>
> ---
>
> **Response**
>
> Thank you for pointing out these relevant works. We have added a discussion of [1, 2, 3, 4] in Appendix J. However, we emphasize that all of these prior methods translate natural language into a **closed-world symbolic PDDL domain** and then rely on **standard off-the-shelf PDDL planners**. In contrast, our work introduces a new planning paradigm: a hybrid natural-language extension of PDDL that enables **open-world reasoning and entailment during planning**.
>
> Concretely, existing approaches:
>
> 1. rely on an LLM to generate a rigid symbolic PDDL domain from natural language,
> 2. can only produce grounded, closed-world plans with no correctness guarantees,
> 3. assume a fully specified, closed-world domain model, and
> 4. cannot handle cases where the agent’s instructions are misaligned with its action model.
>
> By contrast, our approach:
>
> 1. extends symbolic PDDL to directly support natural-language expressions,
> 2. generates open-world conditional plans that remain valid across multiple possible world configurations,
> 3. performs planning directly in an open-world setting with incomplete domain knowledge, and
> 4. leverages the commonsense knowledge of foundation models to handle action–goal misalignment.
>
> **References**
>
> [1] Miglani, Shivam, and Neil Yorke-Smith. *"NLtoPDDL: One-Shot Learning of PDDL Models from Natural Language Process Manuals."* Proc. of the ICAPS Workshop on Knowledge Engineering for Planning and Scheduling (KEPS). ICAPS, 2020.
>
> [2] Xie, Yaqi, et al. *"Translating Natural Language to Planning Goals with Large-Language Models."* arXiv:2302.05128 (2023).
>
> [3] Smirnov, Pavel, et al. *"Generating Consistent PDDL Domains with Large Language Models."* arXiv:2404.07751 (2024).
>
> [4] Wong, Lionel, et al. *"Learning Adaptive Planning Representations with Natural Language Guidance."* arXiv:2312.08566 (2023).

---

### Author Response · Authors · 2025-12-03
**Review Summary**

We thank the reviewers for their thoughtful comments. We have addressed major concerns and added new experiments and comparisons. The reviewers acknowledged NL-PDDL’s novelty and praised the multi-modal evaluation, however, several key aspects were misunderstood. Below we summarize our core contributions and how the revisions address these points.

---

## Contributions and Highlights

- **Sound open-world planning**
  - NL-PDDL introduces the **first open-world embodied planning framework** that generates sound and verifiable plans under partial observability and incomplete domain knowledge, including **unknown object types, relations, properties, locations, and affordances**.

- **Incorporating sensing into open-world planning**
  - NL-PDDL explicitly reasons about **what the agent needs to sense** and how to act across different possible world states which is a capability **missing from existing embodied AI planners**.

- **Generalization across tasks, domains, and modalities**
  - We demonstrate that NL-PDDL **generalizes across open-world and closed-world settings, and across text and vision modalities**, significantly outperforming baselines while using only a fraction of their token cost.

- **Strong zero-shot performance**
  - NL-PDDL combines a formal lifted regression operator with LLM/VLM-based commonsense entailment, achieving strong performance **without any domain-specific data collection, fine-tuning, or prior symbolic knowledge** beyond an action model.

---

## Summary of Reviewer Feedbacks

### **Comparison with existing approaches** [Reviewers 1 and 3]
Reviewers 1 and 3 requested clearer comparisons to existing approaches. We have added additional comparison and related work in Appendix J, and to clarify:

- **Prompt-based LLM/VLM methods:**
  - NL-PDDL relies only on **predicate-level entailment calls**, rather than full sequence generation making use of LLM reasoning makes planning far more efficient.
  - As shown in Tables 1–2, NL-PDDL consistently **outperforms prompt-based approaches** while using only **one-fifth to one-tenth** of their token cost.

- **Fine-tuned VLM or VLA methods:**
  - Unlike baselines that require extensive domain-specific data and fine-tuning, NL-PDDL operates fully in a **zero-shot setting**.
  - Despite this, it **outperforms fine-tuned ALFWorld agents trained on up to 170,000 expert trajectories** (Table 1).

- **Hybrid PDDL approaches:**
  - Existing approaches compile language into closed-world PDDL domains that **require complete object sets and fully specified relations, and assume perfect alignment** between the action model and natural-language goals.
  - These assumptions **break down in open-world scenarios**.
  - NL-PDDL removes these constraints by operating directly over natural-language predicates with lifted regression and LLM-based entailment.

---

### **Low-level Control and Robotics Evaluation** [Reviewer 2]
Reviewer 2 raised questions about robotic applications and low-level control. We clarify that:

- NL-PDDL focuses on **task-level planning**, where agents are evaluated on high-level plan correctness rather than motor control which is standard practice in embodied AI due to the complexity of hardware evaluation.
- Any NL-PDDL plan can be **directly translated into symbolic PDDL**, which is a common interface used by task-level planners that can be integrated in low-level controls.

---

### **Scalability and Generalizability** [Reviewer 3]
NL-PDDL is scalable and generalizable for two key reasons:

- **Lifted reasoning without exhaustive grounding:**
  - NL-PDDL operates at the **lifted level and avoids enumerating all objects and relations**, which symbolic and hybrid planners require.
  - For example, in a domain with 100 objects, a relation like `canHeat(x, y)` would require 10,000 grounded facts, whereas NL-PDDL needs only **one lifted predicate** during regression.

- **Selective information gathering via regression:**
  - NL-PDDL identifies **only the information relevant to achieving the goal**.
  - The regression operator naturally filters out irrelevant objects and relations, enabling scalability and generalization in open-world tasks that grounded planners cannot handle.

---

## **Additional Experiments and Ablations**
For completeness, we added the following items in the revised manuscript:

- A comparison of the total number of encountered predicates versus predefined action-model predicates (Appendix K).
- An experiment evaluating LLM entailment robustness under varied natural-language expressions (Appendix L).
- Measurements of LLM-call counts and latency (Appendix M).
- Expanded discussion comparing NL-PDDL with existing PDDL-based and hybrid NL to PDDL approaches (Appendix J).

---
This summarizes the key points of our revision. We provided detailed rebuttals for every comment raised by each reviewer,  refer to the full rebuttal for complete explanations and supporting evidence.

---

### Meta-Review · Area_Chair_ctMH · 2025-12-14

**Summary:**

The paper proposes a natural language version of the planning language PDDL (NL-PDDL) and introduced LLM entailment for doing common-sense regression planning with NL-PDDL. In my opinion, this work introduces sufficient novelty and represents useful progress towards open-world planning. Reviewer concerns are partially addressed in the rebuttal. My view is that the contributions in the paper outweighs the remaining concerns.

**Reviewer Concerns:**

Reviewer SAUU did not have serious issues with the work.

Reviewer qrc8 questions the efficiency and scalability of LLM entailment calls. The response from the authors is not entirely satisfactory. However, over the longer term this will become less of an issue as compute capacity increases. The reviewer also commented that there is no clear bridge from symbolic plan to robot execution. The authors responded that NL-PDDL is a task-level planner where the actions can be mapped to lower-level robotic controllers. In my opinion, this is fine for the current stage of research. The reviewer asked about the sufficiency of the predicate set. The authors responded that NL-PDDL does not assume a fixed and predefined predicate set, hence handles this issue better than existing methods.

Reviewer DC4D commented that a prior work, LLM-Regress, already proposed combining lifted regression with LLM for open-world planning. The authors explained that their work supports generic entailment reasoning beyond affordance reasoning used in LLM-Regress. The reviewer also noted that a sharp performance degradation of NL-PDDL (Figure 7) is observed. The authors simply pointed out difference scenarios where NL-PDDL outperforms other methods. This is not really satisfactory -- an investigation on the reason for the sharp performance degradation would help understand the strengths and weaknesses of NL-PDDL. The reviewer asked whether semantic ambiguity or variability in NL descriptions can cause the LLM-based entailment reasoning to produce incorrect unifications. The authors did additional experiments showing that current LLMs are quite reliable for the types of entailment questions that they use for this work.

**Reviewer Scores:**

Reviewer SAUU's score will likely be unchanged.
Reviewer qrc8's concerns are only partially addressed, hence uncertain whether score will be changed.
Reviewer DC4D's concerns are mostly addressed, hence the reviewer may increase the score.

---

### Decision · Program_Chairs · 2026-01-26

Accept (Poster)